# Six Years of IKFS-2 Global Ozone Total Column Measurements

Alexander Polyakov [1,*], Yana Virolainen [1], Georgy Nerobelov [1,2], Dmitry Kozlov [3] and Yury Timofeyev [1]

1. Faculty of Physics, St. Petersburg University, 7/9 Universitetskaya nab., St. Petersburg 199034, Russia
2. SRC RAS—Scientific Research Centre for Ecological Safety of the Russian Academy of Sciences, St. Petersburg 187110, Russia
3. Keldysh Research Center, 8 Onezhskaya Str., Moscow 125438, Russia
* Correspondence: a.v.polyakov@spbu.ru; Tel.: +7-(921)-6306502

**Abstract:** Atmospheric ozone plays an important role in the biosphere's absorbing of dangerous solar UV radiation and its contributions to the Earth's climate. Nowadays, ozone variations are widely monitored by different local and remote sensing methods. Satellite methods can provide data on the global distribution of ozone and its anomalies. In contrast to measurement techniques based on solar radiation measurements, Fourier-transform infrared (FTIR) satellite measurements of thermal radiation provide information, regardless of solar illumination. The global distribution of total ozone columns (TOCs) measured by the IKFS-2 spectrometer aboard the "Meteor M N2" satellite for the period of 2015 to 2020 is presented. The retrieval algorithm uses the artificial neural network (ANN) based on measurements of TOCs by the Aura OMI instrument and the method of principal components for representing IKFS-2 spectral measurements. Latitudinal and seasonal dependencies on the ANN training errors are analyzed and considered as a first approximation of the TOC measurement errors. The TOCs derived by the IKFS-2 instrument are compared to independent ground-based and satellite data. The average differences between the IKFS-2 data and the independent TOC measurements are up to 2% (IKFS-2 usually slightly underestimates the other data), and the standard deviations of differences (SDDs) vary from 2 to 4%. At the same time, both the analysis of the ANN approximation errors of the OMI data and the comparison of the IKFS-2 results with independent data demonstrate an increase in discrepancies towards the poles. In the spring–winter period, SDDs reach 8% in the Southern and 6% in the Northern Hemisphere. The technique presented can be used to process the IKFS-2 spectral data, and as a result, it can provide global information on the TOCs in the period of 2015–2020, regardless of the solar illumination and the presence of clouds.

**Keywords:** IKFS-2; ozone total column; ozone measurements; outgoing thermal radiation

## 1. Introduction

Atmospheric ozone plays a dual role for the biosphere: stratospheric ozone absorbs harmful solar UV radiation, whereas tropospheric ozone is a greenhouse gas and a highly reactive pollutant.

In the second half of the 20th century, emissions of man-made ozone-depleting substances (ODSs), especially chlorofluorocarbons, resulted in the regular appearance of ozone holes over the Antarctic region [1]. The decrease in the amount of ozone molecules leads to a significant increase in UV radiation affecting the dynamics and temperature structure of the atmosphere [2].

After the implementation of the Montreal Protocol in 1989 and later its amendments and adjustments, the concentrations of ODSs in the atmosphere have been declining. Remote sensing observations indicate an increase of 0.3% decade$^{-1}$ in near-global (60°S–60°N) total ozone columns (TOCs) over the 1996–2020 period, but this trend is not yet statistically significant [3]. Positive trends have been observed in upper-stratospheric ozone at mid-latitude in both hemispheres due to the decrease in ODSs and stratospheric temperatures

driven by the increase in green-house gases. At the same time, multiply observations show a decrease in ozone concentrations in the lower stratosphere, to a larger extent in tropics.

As the polar ozone depletion is mainly caused by the production of ODSs, and as ODS emissions have been essentially eliminated, the Antarctic ozone hole has diminished in size and depth since 2000. Meteorological conditions (temperature and winds) are the second factor that impacts the spring ozone loss in the polar stratosphere, especially in the Northern Hemisphere as the Arctic is more dynamically variable [4]. The unprecedented chemical ozone loss in the Arctic in the spring of 2020 was driven by a very long and cold Arctic stratospheric winter [5,6]. Notwithstanding the decrease in ODSs, models predict that if the temperatures in the polar stratosphere during Arctic winters continue to decrease, a more significant ozone loss in spring will be observed [7].

The uncertainty of ozone trends and processes in different altitudes and latitudes clearly indicates that the development of new techniques and the implementation of new instruments for ozone monitoring are still crucial tasks.

The international monitoring of the global ozone was initiated in the middle of the 20th century. Nowadays, many different approaches and tools are used to monitor the ozone content in the total atmospheric column and at particular vertical levels (e.g., by aircrafts and ozonesondes) from the ground and from space. According to the WMO (World Meteorological Organization) [3], approximately 400 ground-based observatories have been established around the globe since 1957. Due to their high accuracy and temporal resolution, these ground-based measurements, together with space-based remote observations from satellites, contribute substantially to the global monitoring of TOCs and ozone vertical distribution.

There are three main techniques and several satellite instruments that have been actively used to measure TOCs and ozone profiles during the last few years.

The first is a method based on the transparency of the atmosphere. In this method, direct solar radiation is measured by a satellite in an occultation mode and is used to obtain the vertical distribution of the ozone content at heights higher than 5–10 km. One of the instruments on board the SCISAT (Scientific Satellite) satellite that carries out such observations is the Atmospheric Chemistry Experiment Fourier Transform Spectrometer (ACE-FTS) [8].

According to the second method, TOCs as well as ozone vertical profiles are retrieved from solar backscattered radiation measured in two geometries—nadir and limb. There are quite a lot of satellite measurement systems that provide such observations—the Ozone Monitoring Instrument (OMI, Aura satellite) [9]; the TROPOspheric Monitoring Instrument (TROPOMI, Sentinel-5 Precursor satellite) [10]; the Global Ozone Monitoring Experiment (GOME-2, MetOp satellites) [11]; and the Ozone Mapping and Profiler Suite (OMPS, Suomi and NOAA-20 satellites) [12].

Finally, the third method for TOC and ozone profile retrievals is based on measurements of the outgoing Earth thermal radiation (IR), also in two geometries. Such observations are provided by the following satellite observation systems: the Atmospheric Infrared Sounder (AIRS, Aqua satellite) [13]; the Tropospheric Emissions Spectrometer (TES, Aura satellite) [14]; the Infrared Atmospheric Sounding Interferometer (IASI, MetOp satellites) [15]; the Infrared Fourier Spectrometer-2 (IKFS-2, Meteor-M N2 satellite) [16]; and the Microwave Limb Sounder (MLS, Aura satellite) [17].

The second observational method is the most reliable in the sense of its accuracy (~1–2% TOC retrieval errors). In addition, the OMI and TROPOMI observation systems possess the highest spatial resolution among other ozone-measuring satellites. Spatial resolution constitutes 13 × 24 km for OMI [18–20] and 3.5 × 7 km for TROPOMI [21]. However, observations from this method can be carried out only in the presence of solar illumination (i.e., during daytime). This fact makes this method unusable during polar nights. The third method does not depend directly on incoming solar radiation and can be used for TOCs and ozone profile retrieving even during nighttime. The observational

systems that are based on this method have a relatively high spatial resolution (tens of kilometers), with retrieval inaccuracies of 2–5% [16,22–26].

The quality of the satellite data is controlled on a regular basis by validation using ground-based ozone measurements from the different networks and independent satellite observations.

Garane et al. [21] estimated the quality of TROPOMI TOC measurements by their comparison against daily ground-based observations. The ground-based data consisted of Brewer, Dobson, and DOAS (differential optical absorption spectroscopy) measurements. For these data pairs, the bias totaled 0–1.5%, and standard deviation of differences (SDDs) constituted 2.5–4.5%. In this study, TROPOMI TOC measurements were also compared to OMPS and GOME-2 satellite data. The bias and SDD values obtained were less than 0.7% and 1%, respectively.

Levelt et al. [9] validated OMI TOC measurements for a period of more than 10 years against Brewer and Dobson ground-based data. They demonstrated no drift in differences between satellite and ground-based observations for the whole period. They also showed a small trend in the OMI data versus the TOCs from the SBUV/2 (Solar Backscatter Ultraviolet Radiometer) instrument (NOAA-19 satellite) of about 0.4% per decade, and they showed a bias of −0.9%. A larger (up to 2%) bias was found between the OMI data and the GOME-type TOC measurements.

Virolainen et al. [27] demonstrated seasonal behavior (with a 1.5% amplitude) with regard to differences between OMI TOCs and ground-based TOCs derived by the M-124 filter radiometer in the vicinity of St. Petersburg for the 2009–2012 period.

According to [28], the mean differences and SDDs between several satellite data products (IKFS-2, OMI, TROPOMI) and ground-based standard measurements (Dobson) near St. Petersburg for different periods in the timespan of 2015–2020 constituted 1.8–2.4 and 3.3–3.7%.

Polyakov et al. [29] validated IKFS-2 TOC measurements against Brewer and Dobson hourly (individual) measurements presented at the WOUDC (World Ozone and Ultraviolet Data Center) observational network for the 2019–2020 period. They showed that for direct sun ground-based measurements with a 1 h temporal difference and 70 km spatial differences, the mean bias totaled −1.46% with a SDD of 2.57%.

In the current study, we extended the period of IKFS-2 measurements to 2015–2020, optimized the retrieval algorithm, validated the extended database of IKFS-2 TOC measurements against various ground-based and satellite data, and, finally, we demonstrated the capability and advantages of the IKFS-2 instrument for studying ozone distribution and its anomalies on various spatial and temporal scales.

Section 2 (Materials and Methods) gives an overview of the IKFS-2 instrument, describes the retrieval technique, and summarizes previous studies devoted to IKFS-2 measurements of TOCs. Section 3 (Results) depicts the analysis of the direct differences between IKFS-2 TOC data and independent measurements. Section 4 (Discussion) presents the spatial and temporal analysis of the averaged IKFS-2 TOC data together with other satellite results and demonstrates a few examples of a global ozone distribution analysis based only on IKFS-2 data. Finally, Section 5 (Conclusions) summarizes the results and conclusions of the paper.

## 2. Materials and Methods

### 2.1. The IKFS-2 Instrument

IKFS-2 (Infra Krasny Fourier Spectrometer, where "Krasny" in Russian means "Red") is a Fourier spectrometer [30] with a non-apodized spectral resolution of 0.5 cm$^{-1}$ and is one of the payloads onboard the "Meteor-M N2" series satellite. The instrument and the examples of its usage are described in a review [16]. IKFS-2 records the spectra of outgoing thermal radiation of the Earth's atmosphere and surface. The "Meteor-M N2" satellites operate in sun-synchronous orbits at an altitude of 820 km with an orbital inclination of 98.77° and with equatorial crossing time of 09:30 asc. Regular measurements of spectra

with IKFS-2 were started in March 2015 with the swath width across the orbit of 1000 km and were carried out until December 2020. Therefore, at latitudes from 50°S to 50°N, gaps remained between adjacent orbits, which were filled in on the following days of measurements; in areas with a radius of about 400 km around the poles, measurements were never performed. Thereafter, the swath width was expanded to up to 1500 km to cover the whole territory of the Russian Federation. In this paper, the study of the TOC measurement for the period between Mar 2015 and Nov 2020 is presented, i.e., the IKFS-2 data related to a swath width of 1000 km. Table 1 depicts some principal features of the instrument.

**Table 1.** Basic IKFS-2 parameters.

| Parameter | Requirement |
|---|---|
| spectral range | 5–15 μm (660–2000 cm$^{-1}$) |
| non-apodized spectral resolution | 0.4 cm$^{-1}$ |
| radiometric calibration error (λ = 11 … 12 μm, T = 280 … 300 K), no more than | 0.5 K |
| noise equivalent spectral radiance NESR, W/(m$^2$ sr cm$^{-1}$) | $3.5 \times 10^{-4}$, λ = 6 μm $1.5 \times 10^{-4}$, λ = 13 μm $4.5 \times 10^{-4}$, λ = 15 μm |
| instantaneous field of view (IFOV) | 40 mrad (35 km) |
| swath width | 1000 … 2500 km |
| spatial step | 60 … 110 km |
| sampling period | 0.6 s |

The important parameters in our study are spectral region, spectral resolution, and noise. Table 1 shows that spectral region includes the 15 μm $CO_2$ band, the transparent window, and the 9.6 μm ozone band. The spectral resolution of IKFS-2 allows us to select several spectral channels in different parts of the bands. The measured spectrum comprises 2701 spectral channels with an apodized spectral resolution of 0.7 cm$^{-1}$ in the 660–1210 cm$^{-1}$ spectral region, and of 1.4 cm$^{-1}$ in the 1210–2000 cm$^{-1}$ region. The instrument has only the CdHgTe sensor; thus, its sensibility decreases in a shortwave region. To compensate for this decrease, instrument spectral resolution in the shortwave region was increased.

### 2.2. Ozone Retrieval with the Updated ANN Algorithm

This paper continues a series of studies [16,24–26,29,31] devoted to the determination of TOCs from the IKFS-2 spectra using the artificial neural network technique (ANN). The first study [24] was based on a small dataset that covers the period between March and November 2015. Results of these studies demonstrate the possibility of using the IKFS-2 spectra and the ANN approach to estimate TOCs globally. In [24], the input set of technique parameters consisted of a zenith angle of the satellite and two sets of the principal components (PCs) of the measured spectra: 25 PCs of the whole spectra (660–2000 cm$^{-1}$) and 50 PCs of the ozone absorption band (1000–1200 cm$^{-1}$). PCs are the coefficients of decomposition of a spectrum by empirical orthogonal functions (EOFs), i.e., eigenvectors of the radiance covariance matrix. Later, the authors of [25] showed that the same technique allows for the retrieval of TOCs under cloudy conditions with the same precision. In [26], the measurement period was expanded to up to 2 years (August 2015–July 2017). To calculate the EOFs and PCs of the ozone absorption band, the authors of [26] used the 980–1080 cm$^{-1}$ spectral region that intersects with the ozone absorption band and the 660–1210 cm$^{-1}$ range to calculate the EOFs and PCs of the "whole spectrum". The spectra for the ANN training were derived from a 12 h period of measurements twice a month after the prophylactic service of the instrument. Finally, the sample data for the training consisted of $2.9 \times 10^6$ spectra. To reduce the size of the data sample, the authors selected about $10^6$ pairs of "IKSF spectrum–OMI TOC" using pseudo random selection. To study TOC anomalies (mini-holes), a continuous sample dataset of IKFS-2 spectra measured between

3 Oct 2015 and 30 Apr 2016 was also used. A dataset size constituted approximately $2 \times 10^7$ spectra.

The main advantage of the IR technique over OMI-like instruments that use backscattered solar radiation to retrieve TOCs is the possibility of performing measurements during polar nights. In [31], the authors used IKFS-2 TOCs to analyze the polar ozonosphere for the 2015–2016 winter. Finally, in [29], the ANN was applied to the 2019–2020 period. The authors showed the impropriety of applying the ANN trained with the 2015–2017 dataset to the 2019–2020 period. The SDDs between the satellite and ground-based TOC measurement results rose to 12% for the 2019–2020 period instead of 3–5% for the 2015–2017 period. The ANN was retrained with a random selection of 10% of the 2019–2020 dataset, and the SDDs returned to their previous values. Additionally, the dependence of differences between the satellite (IKFS-2) and ground-based (Brewer and Dobson instruments) hourly measurement data on spatial and temporal mismatch was studied. The values of the mismatch between satellite and ground-based observations were optimized to 70 km and 1 h. The decrease in mismatched values did not improve the agreement between the satellite and ground-based data, but the increase in mismatches worsened the agreement and increased the SDDs. The best results were shown for the comparison of IKFS-2 TOC to ground-based TOC measurements using direct solar radiation. For the 70 km and 1 h mismatches, the mean bias constituted $-1.46\%$ with a SDD of 2.57%. IKFS-2 underestimated the TOCs with respect to OMI by 0.1–1.0%. Moreover, in that paper, the IKFS-2 TOCs were used to study the features of TOC spatio-temporal variability during the 2019–2020 Arctic winter.

As in the enumerated papers, we used one of the simple three-level perceptron ANN. Activation function $f$ is a hyperbolic tangent. Figure 1 shows a schematic picture of the ANN, the detailed description of which is presented in Supplementary Materials.

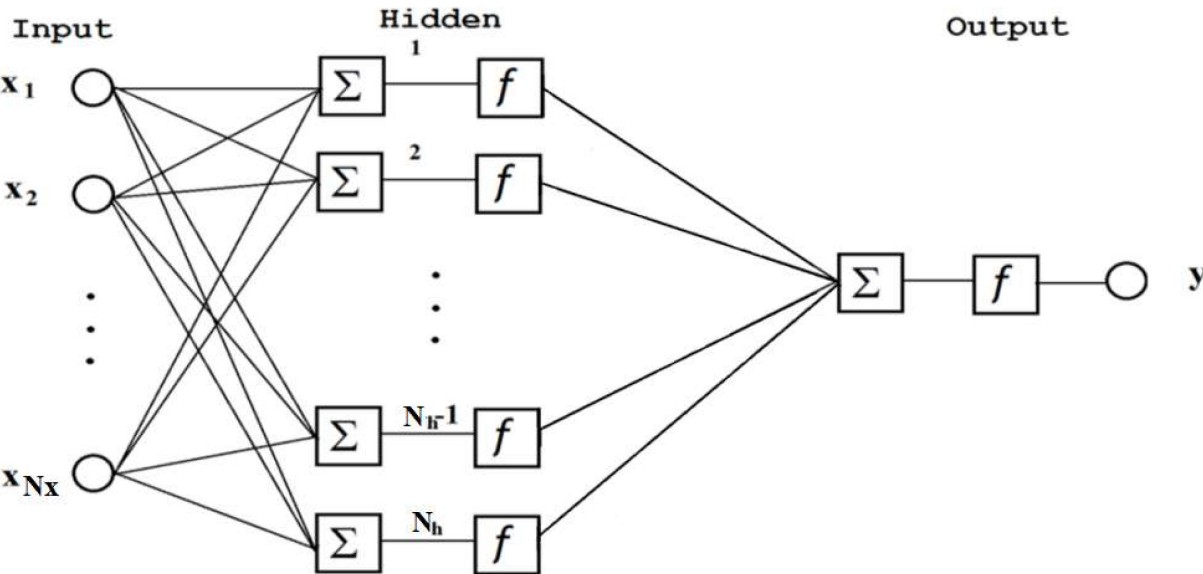

**Figure 1.** The structure of the ANN used to solve the inverse problem of TOC estimation. Variables $x_i$ are input parameters, $f$ is an activation function, $\Sigma$ means "summator", and $y$ is a result.

The input set of parameters (predictors) of the ANN $\vec{X}$ includes a zenith angle of the satellite, which is derived from the observed pixel of the Earth's surface and two sets of spectral PCs. The first set consists of $N_{PC_{total}}$ PCs in the 600–1210 cm$^{-1}$ spectral region. This region includes the $CO_2$ spectral absorption band that contains information on atmospheric temperature profile, the transparency window with the information on the surface, and the 9.6 μm ozone absorption spectral band. The second set consists of $N_{PC_{O3}}$ PCs in the 980–1080 cm$^{-1}$ spectral region of the ozone absorption band. In contrast to previous studies, we introduce two additional predictors: the latitude of measurement pixel and the fraction

of a year, i.e., the number of a day divided by the number of days in the year. Thus, total number of the $N_x$ predictors equals:

$$N_x = S + N_{PC_{total}} + N_{PC_{O3}} \tag{1}$$

In Equation (1), $S = 3$ when the latitude and the fraction of year are used. Then, total number of coefficients $N$, which is fitted into the process of ANN training, is equal to $N = N_x N_h + 2N_h + 1$, where $\mathrm{N_h}$ is the number of neurons of a hidden layer (see Figure 1).

The ANN training data set is based on TOCs measured by the OMI instrument aboard the Aura satellite [19]. OMI measures backscattered solar radiation in the 270–500 nm wavelength range. Its swath width is 2500 km, its spectral resolution is about 0.5 nm, and its horizontal resolution constitutes $13 \times 24$ km$^2$. The OMI TOCs measurement errors are within a range of 1–2% [18,19]. We use the OMI measurements due to their high precision and global coverage.

High-precision ground-based measurements are performed in a limited set of locations; therefore, they cannot be used for ANN training, whereas satellite measurements provide a global distribution of TOCs measured at various atmospheric states. Calibration of IKFS-2 TOC measurements by OMI TOCs in the ANN algorithm brings to IKFS-2 TOC errors OMI TOC measurement errors and errors due to the spatial and temporal differences in IKFS-2 and OMI measurements only. OMI TOC data are the data with known accuracy that were tested in a series of comparisons with independent data (see, e.g., [9,19,32]).

The use of a complex ANN with many layers and coefficients can reduce the approximation error of the training dataset, but possibly at the cost of the parasitic accounting of errors in these data. Simplifying the ANN structure and minimizing the number of fitted coefficients make it possible to construct a solution operator that describes the physical relationships between predictors and TOCs. In addition, it allows us to avoid the parasitic accounting of various errors contained in the training dataset. The time spent for ANN training does not cost much. Earlier in the papers [24–26,29], the 25—50—30 ($N_{PC_{total}}$—$N_{PC_{O3}}$—$N_h$) scheme was chosen. In this paper, the 30—60—50 ANN scheme is additionally considered. In Table 2, we present the approximation errors (loss function) and results of the comparison against independent data for these two ANNs. Row 2 of Table 2 shows that the new scheme noticeably reduces the TOC approximation error by 0.31 DU. However, the main criterion for the choice of the technique for solving the inverse problem should be the agreement of the retrieved TOCS with the most accurate independent measurements. These data are ground-based measurements by Dobson and Brewer instruments. In the current study, we retrieved TOCs from IKFS-2 spectra for 6 years (2015–2020) using two ANNs and compared the retrievals with hourly TOC ground-based measurements. The details of these comparisons are described below in Section 3. The results of the comparisons are shown in Table 2 in columns "WOUDC" and "EUBREWNET". In addition, Table 2 also shows the results of the comparison between IKFS-2 and TROPOMI data, which has been onboard the Sentinel 5P satellite [10] since May 2018 (see details in Section 4).

**Table 2.** The training errors (loss function) of two ANNs and the global mean differences between the IKFS-2 TOC and independent data of ground-based and satellite measurements.

| N | NPC $_{total}$ | NPC $_{O3}$ | $N_h$ | N Param. | Approximation Error, DU. | WOUDC | | EUBREWNET | | TROPOMI | |
|---|---|---|---|---|---|---|---|---|---|---|---|
| | | | | | | Bias, % | SD, % | Bias, % | SD, % | Bias, % | SD, % |
| 1 | 25 | 50 | 30 | 2401 | 8.36 | −0.23 | 2.9 | −0.40 | 2.7 | −1.2 | 3.1 |
| 2 | 30 | 60 | 50 | 4751 | 8.05 | −0.25 | 2.8 | −0.39 | 2.8 | −1.3 | 3.0 |

Both ANNs are in close agreement with the independent data. The difference between the two ANNs is negligibly small. Thus, we can choose the ANN with the minimal number of fitted coefficients. In this study, all TOCs were retrieved with the following set of

predictors: zenith angle, latitude, fraction of year, 25 PCs of broad spectra, and 50 PCs of ozone absorption band. A total of 30 hidden level neurons were used. Therefore, the number of PCs is the same as in previous papers—the difference lies in the use of latitude and year fraction only.

## 3. Results

A necessary step in the development of a technique for processing spectral data is the validation of the results based on a comparison with independent data. The results of these comparisons are given in the next sections. We use two main parameters to describe the differences: the average relative difference, bias, and the standard deviation of the relative difference, SDD, calculated using relations (2):

$$Bias = \frac{100}{n} \sum_{i=1,n} \frac{(U_i - W_i)}{W_i}, \; SDD = 100 \sqrt{\sum_{i=1,n} \left( \frac{U_i - W_i - Bias}{W_i} \right)^2 / (n-1)}, \quad (2)$$

where $U_i$ is the TOCs measured by IKFS-2, $W_i$ is the independently measured TOCs, $i$ and is the serial number of a pair of two kinds of measurements. Bias and SDD are given in percentages with respect to independent observation data.

### 3.1. Comparison versus Ground-Based Measurements

To validate the results of satellite total column ozone measurements, data from ground-based ozone networks based on Dobson and Brewer instruments are often used. For the validation of the IKFS-2 TOCs, only direct solar observations are used as a reference dataset, as they are the most accurate. The accuracy of direct-sun TOC individual measurements totals 1–2% [33].

Boynard et al. [15,34], Garane et al. [21], McPeters et al. [18], and Orfanoz-Cheuquelaf et al. [35] used daily TOCs derived from direct solar measurements at WOUDC sites. Additionally, Garane et al. [21] compared TROPOMI ozone data to hourly measurements from the Canadian Brewer Network and the European Brewer Network—Eubrewnet. In the current study, we compared IKFS-2 TOC data to individual (hourly) Brewer and Dobson measurements from the WOUDC and Eubrewnet networks.

The choice of sites and data pairs was based on a spatio-temporal match with the IKFS-2 TOC measurements. Figure 2 depicts the locations of the ground-based sites chosen. Some of these sites provide only daily values of TOCs, some provide only hourly values, and others provide both types of data.

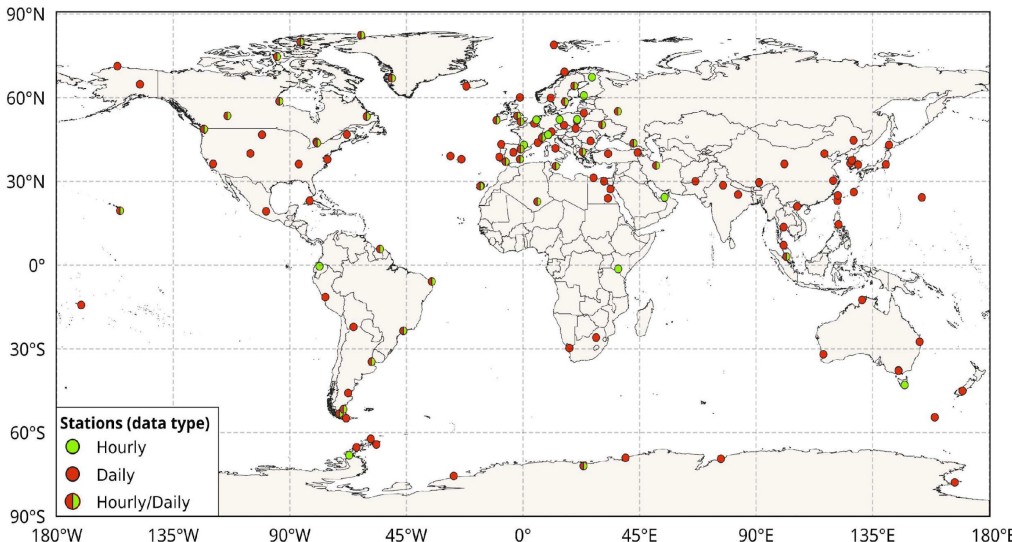

**Figure 2.** Locations of ground-based observational sites of WOUDC and Eubrewnet networks.

Figure 2 demonstrates that the distribution of observational sites is extremely non-uniform. Most sites are located in the Northern Hemisphere, mainly in Europe. A few high-latitude sites in both the Northern and Southern Hemispheres are of great interest due to the localization of the polar stratospheric vortex (PSV) zones and ozone depletion.

For each station, variability of hourly TOC measurements, standard deviation (SD), an amplitude (a difference between maximum and minimum values) for each day, root-mean-square (RMS) of SDs, maximum SD and amplitude were calculated and analyzed. For Eubrewnet stations, the RMS of the SDs is between 2.6 DU (station 002, Tamanrasset) and 17.3 DU (447, Goddard Space Flight Center); maximum SDs are between 7.0 DU (997, Lampedusa) and 127.2 DU (309, Copenhagen). For the WOUDC stations, the RMS of the SDs is in the range of 2.0 DU (200, Cachoeira-Paulista) to 28.8 DU (035, Arosa); maximal SDs vary from 3.8 DU (200, Cachoeira-Paulista) to 168.4 DU (035, Arosa) and 167.9 DU (479, Aosta). The RMS of the SDs is mainly within 5–10 DU, and maximal SDs are within several tens of DU. For the WOUDC data, maximal amplitudes usually exceed 100–200 DU. For Eubrewnet data, the amplitudes are about 100 DU.

These results demonstrate the importance of the comparison of satellite TOCs with ground-based hourly data. In [29], it was pointed out that the SD of the differences between satellite and ground-based TOCs significantly increase for a temporal data mismatch larger than one hour.

In [15,34], the authors used the coincidence criteria of a 50 km radius between the IASI center of the pixel and the geolocation of the ground-based stations. Garane et al. [21] studied the effect of co-location and temporal variability between TROPOMI and ground-based (Dobson and Brewer) TOC measurements. They kept the co-location criterion for satellite data validation at 10 km and temporal criterion at not more than 40 min. Taking into account the difference in pixel size of the TROPOMI (7 km) and IKFS-2 (35 km) instruments, we conclude that the criteria of coincidence for validation between the satellite and ground-based measurements (70 km and 1 h) proposed by Polyakov et al. [29] for IKFS-2 agrees well with those in the studies by Garane et al. [21] and Boynard et al. [15].

### 3.1.1. Comparison versus Hourly Dobson and Brewer Data

Table 3 depicts the results of the comparison between IKFS-2 TOC data and hourly measurements by Dobson and Brewer instruments from the WOUDC network. The total pairs number equals 264,395, and bias and SD constitute −0.23% and 2.89%. We excluded from consideration the stations that demonstrate excessively large values of amplitudes or RMS of SD during the day (see section above Section 3.1). We deliberately do not include these stations, since the analysis of data from the WOUDC and Eubrewnet networks is not the topic of this paper. Thus, ground-based data from 21 WOUDC sites are used in the comparisons (see Table 3). Only two of these sites are in the Southern Hemisphere, in the tropics. Most of the sites are located in the midlatitudes of the Northern Hemisphere.

Dobson instruments are considered as the standard against which most other ground-based instruments for total ozone measurements are calibrated. Total discrepancies between Dobson and IKFS-2 TOCs (Bias = 0.1%, SD = 2.04%) are better than when considering Brewer and Dobson instruments together (Bias = −0.8%, SD = 2.9%). Maximal bias is for Mauna Loa site elevated at 3397 m asl (3.2%). It may be due to the variability in the surface altitudes of the IKFS-2 pixels in a radius of less than 70 km around this site (from almost 0 to 4000 m), with predominating altitudes below the site elevation. Obviously, satellite TOC measurements exceed those of ground-based data due to the amount of ozone in the atmospheric layer below than at the station level. Significant positive biases are also observed for the Kislovodsk (1.6%, 2070 m asl) and Lannemezan (2.4%, 590 m asl) stations. On average for all stations, bias is negative, i.e., the IKFS-2 data underestimate the ground-based TOCs. SDDs for Dobson and Brewer together are in the range of 1.2–3.5%. SDD is maximal for the Mauna Loa site due to the surface-level variability of the IKFS-2 TOCs. Besides Mauna Loa site, only the Edmonton site depicts an SDD value higher

than 3%. Therefore, we can conclude that on average, the SDD between the IKFS-2 and ground-based TOCs is less than 3%.

**Table 3.** Differences between IKFS-2 and ground-based TOCs (WOUDC) relative to ground-based measurements; spatial and temporal mismatches are 70 km and 1 h; second column denotes the type of instruments (B—Brewer, D—Dobson).

| N | I | Station | Latitude, Degrees | Longitude, Degrees | Altitude, m | Pairs Number | Bias, % | SDD, % |
|---|---|---|---|---|---|---|---|---|
| 1 | B | Eureka | 80.050 | −86.420 | 9 | 82,767 | −0.2 | 2.6 |
| 2 | B | Resolute | 74.700 | −94.970 | 68 | 11,979 | −0.6 | 2.1 |
| 3 | B | Churchill | 58.750 | −94.070 | 26 | 8360 | −1.6 | 3.0 |
| 4 | B | Obninsk | 55.100 | 36.610 | 100 | 1044 | −0.2 | 2.7 |
| 5 | B | Edmonton | 53.550 | −114.110 | 752 | 8496 | 1.0 | 3.1 |
| 6 | B | Goose Bay | 53.310 | −60.360 | 26 | 11,752 | 0.0 | 2.2 |
| 7 | B | Lindenberg | 52.209 | 14.121 | 127 | 9472 | −1.3 | 2.9 |
| 8 | B | De Bilt | 52.100 | 5.180 | 24 | 12,558 | −2.6 | 2.1 |
| 9 | D | Kyiv-Goloseyev | 50.364 | 30.497 | 206 | 3763 | −0.1 | 1.9 |
| 10 | B | Saturna Island | 48.770 | −123.130 | 202 | 7686 | 0.4 | 2.6 |
| 11 | B | Aosta | 45.740 | 7.360 | 570 | 1022 | 0.3 | 2.0 |
| 12 | B | Egbert | 44.230 | −79.780 | 264 | 7493 | −1.6 | 2.1 |
| 13 | D | Lannemezan | 44.129 | 0.370 | 590 | 131 | 2.4 | 2.0 |
| 14 | B | Toronto | 43.780 | −79.470 | 202 | 50,937 | −1.0 | 2.2 |
| 15 | B | Kislovodsk | 43.730 | 42.660 | 2070 | 3783 | 1.6 | 2.4 |
| 16 | B | Thessaloniki | 40.634 | 22.956 | 60 | 6959 | −1.0 | 2.2 |
| 17 | D | University of Tehran | 35.730 | 51.380 | 1419 | 674 | 1.1 | 2.0 |
| 18 | B | Mauna Loa (HI) | 19.540 | −155.580 | 3397 | 24,581 | 3.2 | 3.5 |
| 19 | B | Paramaribo | 5.806 | −55.210 | 16 | 10,880 | −0.5 | 2.1 |
| 20 | D | Natal | −5.835 | −35.207 | 49 | 32 | 0.5 | 1.2 |
| 21 | D | Cachoeira-Paulista | −22.69 | −46.200 | 574 | 26 | −3.5 | 1.6 |
| | | Total | | | | 344,412 | −0.8 | 2.9 |

Besides WOUDC data, Eubrewnet provides ground-based measurements by Brewer instruments only. The results of the comparison between the IKFS-2 data and the hourly ground-based measurements at the Eubrewnet sites are shown in Table 4. Some of the Eubrewnet sites are within the WOUDC network. Using the same mismatch criteria of 70 km and 1 h and the above-mentioned variability selection for comparison, we considered data from twenty-nine stations, eight of which are located in the Southern Hemisphere, including two stations in the Antarctic continent. For the comparison, 196,929 pairs of measurements were selected. On average, the bias and SDD between the IKFS-2 and ground-based TOC data constitute −0.40% and 2.7%, which does not significantly differ from the results of the comparison with the WOUDC network (−0.8 and 2.9%, respectively).

A maximum bias of 3.2% was found for the Punta Arenas and Obninsk sites. Maximum SDDs of 3.7 and 3.6% were obtained from the Sondrestrom, Río Gallegos, and San Marten stations. Note that there is a trend towards a slight increase in SDD for the stations located south of 50°S.

Finally, we can conclude that for the six years of IKFS-2 TOC measurements, the SDD of the individual Brewer and Dobson ground-based measurements is less than 3%. Considering the level of errors of the ground-based measurements, we may presume that the random errors of the IKFS TOC measurements are about 2.5%.

**Table 4.** Differences between IKFS-2 and ground-based TOCs (Eubrewnet) relative to ground-based measurements; spatial and temporal mismatches are 70 km and 1 h.

| N | Station | Latitude, Degrees | Longitude, Degrees | Altitude, m | Pairs Number | Bias, % | SDD, % |
|---|---------|-------------------|---------------------|-------------|--------------|---------|--------|
| 1 | Sodankyla | 67.368 | 26.633 | 100 | 13,409 | 0.5 | 2.2 |
| 2 | Sondrestrom | 66.996 | −50.621 | 150 | 10,742 | −0.8 | 3.7 |
| 3 | Vindeln | 64.244 | 19.767 | 225 | 12,139 | −1.1 | 3.0 |
| 4 | Jokioinen | 60.814 | 23.499 | 106 | 1041 | −1.6 | 2.7 |
| 5 | Norrkoping | 58.580 | 16.150 | 43 | 16,162 | −0.9 | 2.1 |
| 6 | Obninsk | 55.099 | 36.607 | 100 | 616 | −3.0 | 2.7 |
| 7 | Manchester | 53.470 | −2.230 | 76 | 5871 | −2.5 | 2.3 |
| 8 | Warsaw | 52.246 | 20.940 | 120 | 4841 | −1.9 | 1.9 |
| 9 | Valentia | 51.930 | −10.250 | 14 | 5799 | −2.0 | 2.4 |
| 10 | Reading | 51.440 | −0.940 | 61 | 7958 | −2.7 | 2.7 |
| 11 | Arosa | 46.783 | 9.675 | 1840 | 14,249 | −0.3 | 2.0 |
| 12 | Aosta | 45.742 | 7.357 | 570 | 6823 | 0.5 | 2.6 |
| 13 | Zaragoza | 41.634 | −0.881 | 250 | 5843 | −2.0 | 3.3 |
| 14 | Thessaloniki | 40.634 | 22.956 | 60 | 7503 | −0.3 | 2.5 |
| 15 | Murcia | 38.028 | −1.169 | 69 | 4744 | −2.3 | 1.6 |
| 16 | El Arenosillo | 37.100 | −6.730 | 41 | 9662 | −1.1 | 2.1 |
| 17 | Lampedusa | 35.518 | 12.630 | 50 | 502 | −1.3 | 2.1 |
| 18 | Izana | 28.308 | −16.499 | 2370 | 27,809 | 2.3 | 1.6 |
| 19 | Abu Dhabi | 24.339 | 54.640 | 20 | 3682 | −1.4 | 3.3 |
| 20 | Tamanrasset | 22.790 | 5.529 | 1320 | 12,104 | −1.8 | 2.0 |
| 21 | Petaling Jaya | 3.100 | 101.650 | 46 | 4816 | −1.3 | 1.6 |
| 22 | Izobamba | −0.366 | −78.550 | 3058 | 167 | −1.2 | 2.4 |
| 23 | Nairobi | −1.301 | 36.759 | 1795 | 865 | 0.3 | 2.3 |
| 24 | Buenos Aires | −34.583 | −58.483 | 25 | 364 | −1.6 | 1.3 |
| 25 | Hobart | −42.904 | 147.327 | 20 | 12,462 | 0.0 | 2.7 |
| 26 | Rio Gallegos | −51.601 | −69.319 | 5 | 2390 | −1.0 | 3.6 |
| 27 | Punta Arenas | −53.137 | −70.880 | 22 | 3895 | 3.3 | 3.4 |
| 28 | San Marten | −68.130 | −67.106 | 30 | 126 | 2.2 | 3.6 |
| 29 | Princess lisabeth | −71.950 | 23.350 | 1390 | 345 | −0.8 | 3.0 |
|  | Total |  |  |  | 196,728 | −0.40 | 2.7 |

### 3.1.2. Comparison versus Daily Dobson and Brewer Data

The number of stations that provided hourly TOC data was not sufficient to analyze the latitudinal dependence of the TOC differences by different measurement methods. A larger number of stations in the WOUDC network provide daily TOC data. Thus, ground-based TOC measurements from 191 stations were compared to the IKFS-2 data to analyze the latitudinal dependence of the differences between datasets. To estimate an optimal spatial mismatch between the IKFS-2 and the ground-based data, we analyzed the dependence of the SDD RMS on the mismatch values. It was found that the RMSs of the SDD increased with mismatches greater than 300 km. In the current study, a 150 km spatial mismatch was used, and IKFS-2 data were selected for the same dates of the ground-based measurements (in UTC). For the daily data comparison, global averaged bias equals −0.59%, and SDD equals 4.5%. To analyze the latitudinal and seasonal dependences of the differences, biases and SDDs were calculated in latitude bands with a 10-degree width for each season (Figure 3).

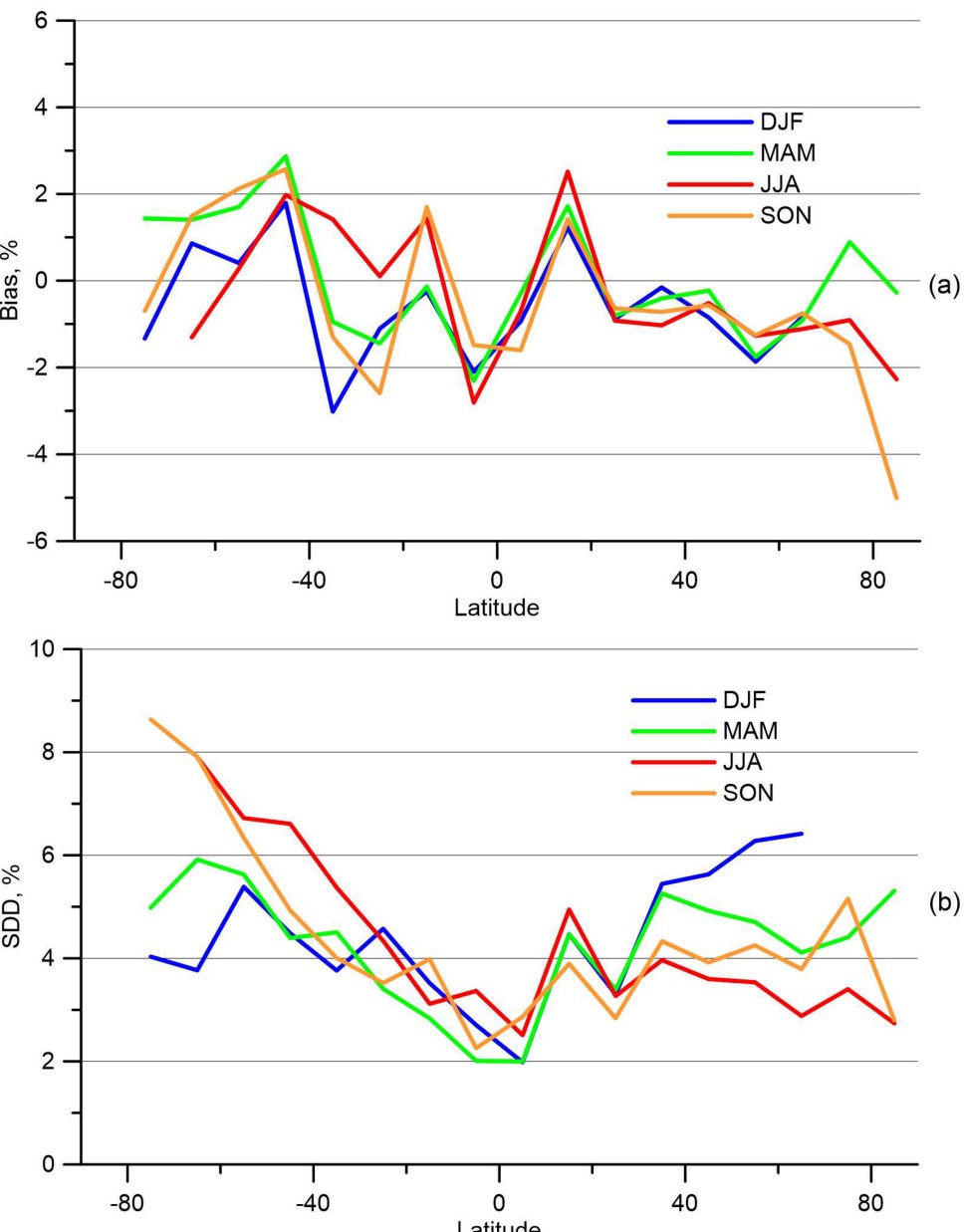

**Figure 3.** Latitudinal and seasonal variation of TOC bias (**a**) and SDD (**b**) between IKFS-2 and ground-based WOUDC daily measurements relative to the ground-based data; DJF—December, January, February; MAM—March, April, May; JJA—June, July, August; SON—September, October, November; GB—ground-based.

Figure 3a shows the bias between the IKFS-2 and the daily ground-based direct-sun TOC data. Most of values are less than 2–3%, except for during autumn in the Arctic, where the bias equals −5%. Such a large bias can be explained by the fact that during the SON period, the comparison between the datasets in the 80°–90°N latitudinal range was performed using the data for the Alert site only and in early September (only 18 days) with low sun elevation. The maximum elevation of the Sun in these days varied from 10° to 15°. Therefore, the trajectory of solar radiation crossed the maximum height of the ozone at distances of about 200–300 km and more so to the south of the station. Verhoelst et al. [36] analyzed the error budget by comparing satellite and ground-based measurements. The sampling difference errors exceeded the measurement uncertainties by 10% and more so in some extreme cases at most mid- and high-latitude sites. Such extreme cases

can be observed, for example, when high a TOC gradient is induced by a stratospheric polar vortex.

In the 20°–60°N region, the bias is stable and is less than 2%. Most of the stations are concentrated in these latitudes (see Figure 2), and the biases of the individual stations are smoothed due to averaging. In other latitudinal regions, the density of the stations is much lower, and the biases between the IKFS-2 and the ground-based TOC data can be affected by local site features or instrument calibration error. For example, the Mauna Loa station is located at a latitude of 19.53°N, and due to differences in surface elevation, it has a large bias of 3.5% (see above Section 3.2). Many (24,765) comparisons were made for this station, resulting in a bias of about 2% in the 10–20 N region for all seasons.

SDDs (Figure 3b) are lowest in tropical latitudinal regions (20°S–20°N), except for the mentioned Mauna Loa site. In addition, SDDs increase towards the poles, which are most noticeable in winter and spring, especially in the Southern Hemisphere. This growth can be caused by several reasons:

- Displacement of the intersection between the solar radiation trajectory and the layer of maximum ozone content from the location of the station due to the low Sun;
- Greater ozone variability in polar latitudes, both in space and time, compared to the tropical regions;
- An increase in IKFS-2 TOC retrieval errors that is associated with a possible decrease in the altitude gradient of the air temperature in the polar atmosphere and low surface temperature.

### *3.2. Comparison versus Satellite Data*

Although ground-based measurements are the most accurate, the small number of stations and their fixed location limit the validation of IKFS-2 data on a global scale. Satellite data provide much better spatial coverage. Nowadays, TOC data from several satellite measuring programs are available; the most precise of which are OMI and TROPOMI.

The comparison of satellite Level 2 data (i.e., individual measurements) provides the most accurate error estimation. Below, we present a comparison of global TOCs between those from IKFS-2 and two other satellite instruments—OMI and TROPOMI.

### 3.2.1. Comparison versus OMI Data—Approximation Errors

Note that OMI data are used for ANN training. Hence, the comparison between OMI and IKFS-2 data can provide an estimation of the approximation errors of the ANN training sample or the training errors. However, in addition, the comparison with OMI TOCs allows us to estimate the latitudinal and seasonal dependences of the IKFS-2 TOC measurement errors. Figure 4 depicts the differences between the IKFS-2 and the OMI TOC dataset that was used for the ANN training. Due to the use of OMI TOCs for the ANN training, the overall bias is almost 0%. Nevertheless, the differences for latitudinal regions and seasons show non-zero values. Figure 4a depicts that the biases (in absolute values) do not exceed 0.5% in the Northern Hemisphere and increase to 1.5% near the South Pole in spring (negative bias). The sign of the bias is changing from each of selected zones to the next. The SDDs (Figure 4b) are also largest near the South Pole and reach 7.5% in winter and 6.5% in spring. A high-altitude snowy surface and a cold troposphere cause a decrease in IR measurement informativity and, consequently, an increase in differences for the South Pole region. The SDDs in the Northern Hemisphere are less than 3%, except for winter, when they are about 4.5%. In the tropics, the SDDs constitute approximately 2%. Overall SDD value between the TOCs by IKFS-2 and OMI constitutes 2.75%.

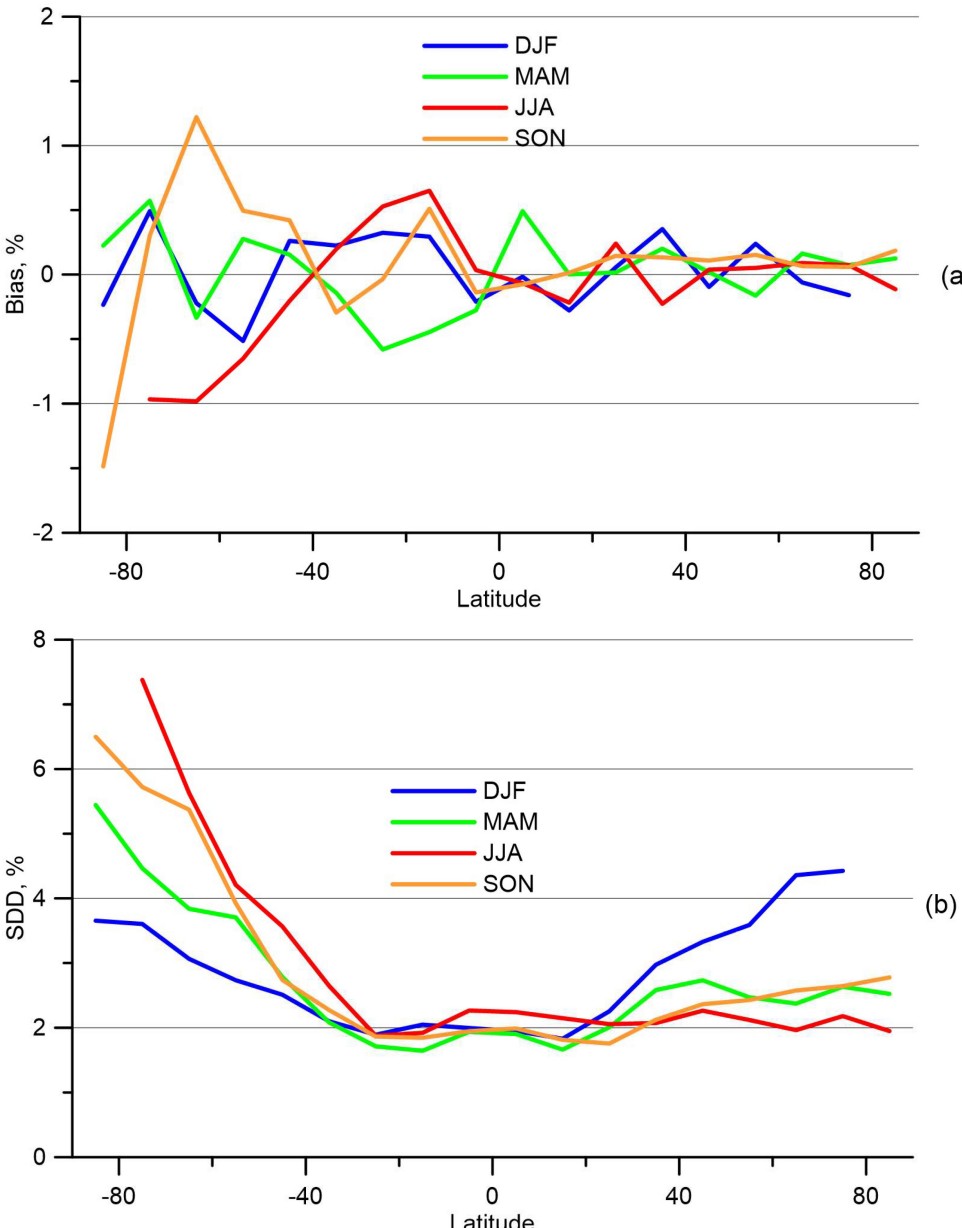

**Figure 4.** Latitudinal and seasonal variations of TOC bias (**a**) and SDD (**b**) between IKFS-2 and OMI data relative to the OMI measurements; DJF—December, January, February; MAM—March, April, May; JJA—June, July, August; SON—September, October, November.

Despite the increase in errors for the polar regions, IKFS-2 provides information on the TOC variability during the polar night season when TOC measurements based on solar radiation are impossible. Figure 5a,b demonstrates the TOC spatio-temporal distribution around the South Pole from IKFS-2 (top) and OMI (bottom) measurements in 22–25 July 2020 and 12–15 August 2017, respectively. Both periods were during a polar night. TOC distribution by IKFS-2 depicts low ozone content near the South Pole, while OMI demonstrates large gaps in the data due to solar radiation absence during a polar night.

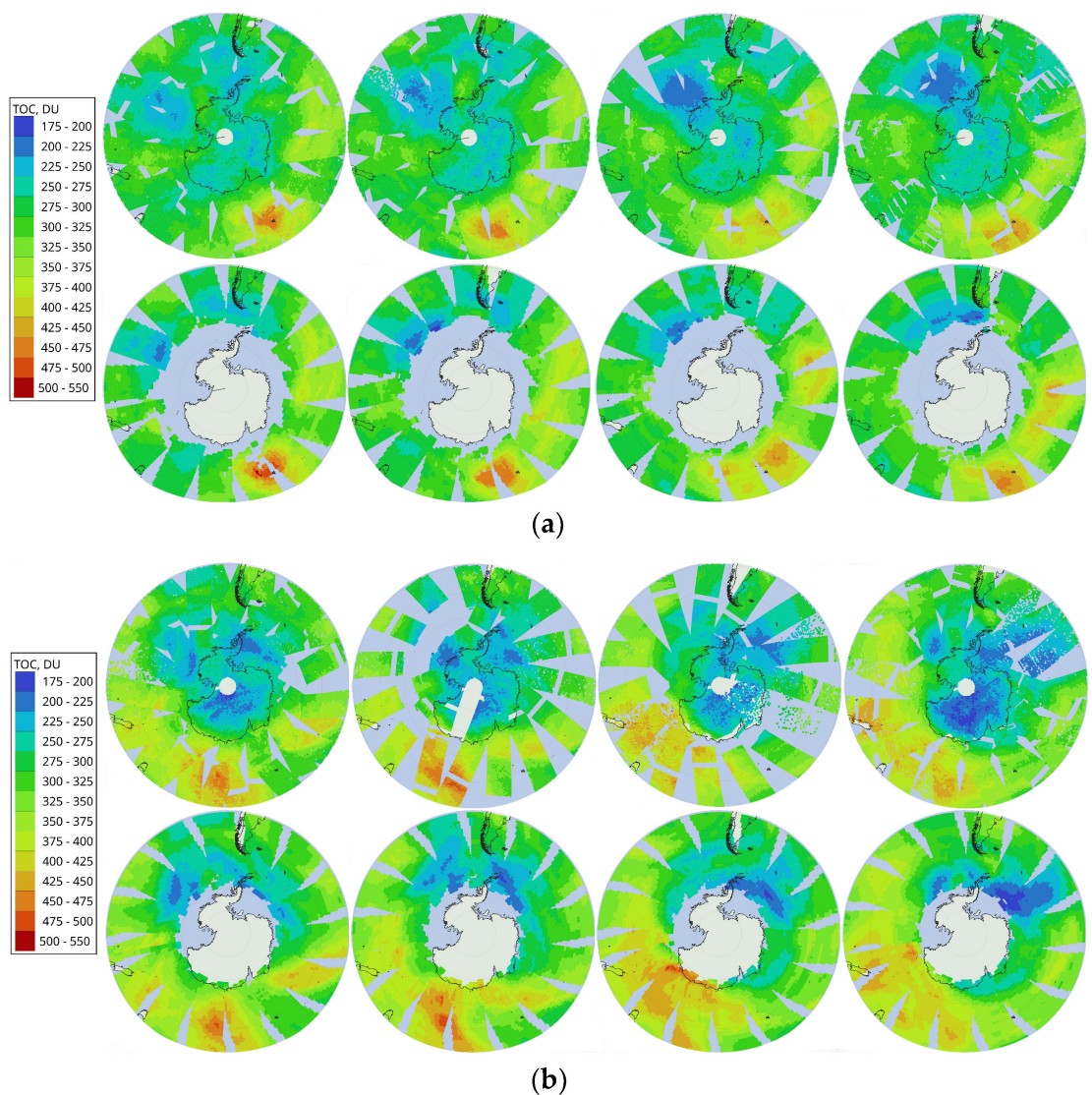

**Figure 5.** TOCs spatio-temporal distribution around the South Pole by IKSF-2 (**top**) and OMI (**bottom**) data; (**a**)—22–25 July 2020; (**b**)—12–15 August 2017, (from **left** to **right**).

### 3.2.2. Comparison versus TROPOMI

The Copernicus S5P satellite with a TROPOMI instrument onboard [10] was launched on 13 October 2017. S5P is the first of the Sentinel satellites planned to measure the composition of the atmosphere for a least a seven-year mission (http://www.tropomi.eu/, accessed on 5 March 2023). For the comparisons with the IKFS-2 TOC data, we used the Level 2 TROPOMI measurements dating back to May 2018. The spatial resolution of TROPOMI is less than 10 km (WMO OSCAR database). The TROPOMI data was filtered by a quality flag (greater than 0.9). Due to the distinctive features of the Meteor-M N2 and S5P satellite orbits, we used TOC data with a temporal mismatch of 6 h. Smaller mismatches limit data availability in the tropical and mid latitudes. To exclude unreliable near-zero TOC values, we considered for comparison the TROPOMI TOC data in the 100–650 DU range. As a result, the number of data pairs compared is ~$1.4 \times 10^9$. The global mean bias between the IKFS-2 and TROPOMI data relative to TROPOMI constitutes $-1.2\%$ with an SDD of 2.75%. The zonal and seasonal dependencies of the data differences (biases and SSDs) are shown in Figure 6.

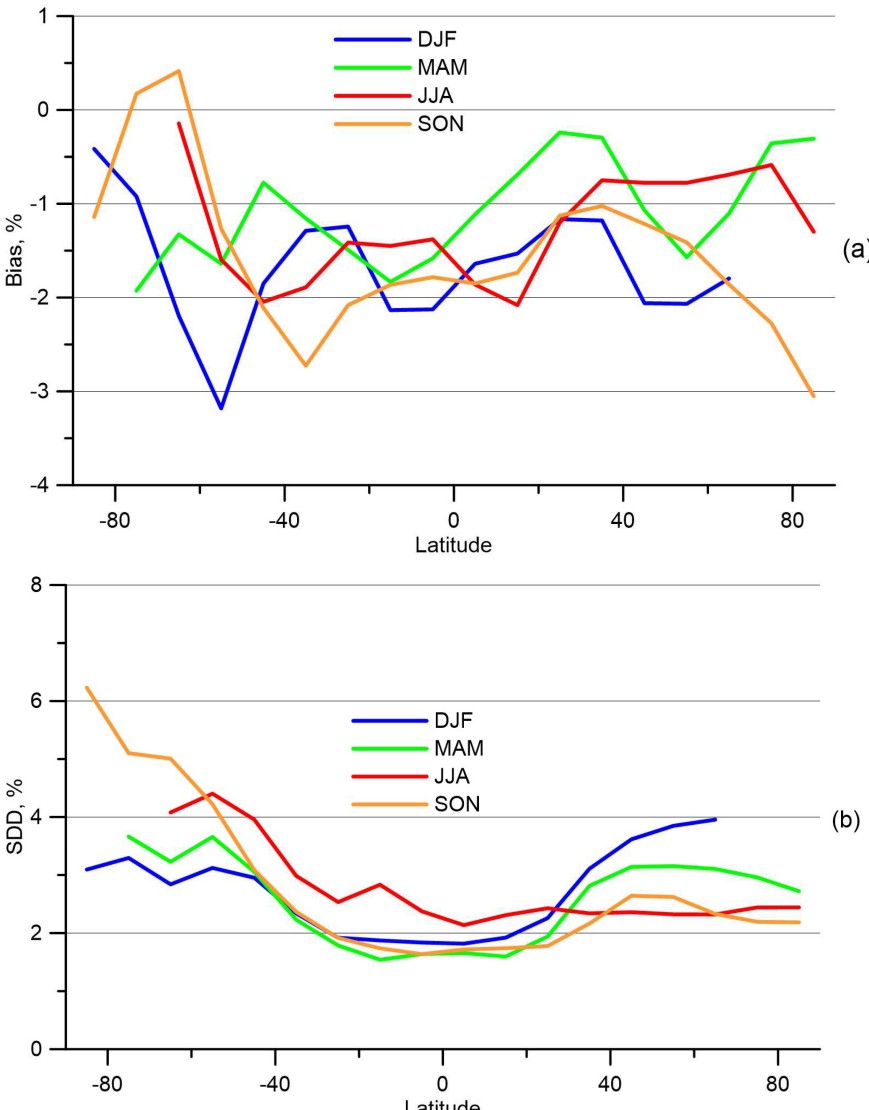

**Figure 6.** Latitudinal and seasonal variations of TOC bias (**a**) and SDD (**b**) between IKFS-2 and TROPOMI data relative to TROPOMI measurements; DJF—December, January, February; MAM—March, April, May; JJA—June, July, August; SON—September, October, November.

Like the comparison with OMI, the best fits between the IKFS-2 and the TROPOMI data were found in the tropical region, with increasing discrepancies towards the poles (Figure 6). However, the biases are mainly negative, except for the 60°–80°S latitudinal region. A possible reason for that may be a shift between the TROPOMI and the OMI data. The SDDs are less than in case of OMI, and the maximum SDD is ~6%. Finally, IKFS-2 and TROPOMI TOCs closely agree, and the discrepancy between the data pairs probably does not exceed a sum-of-errors estimation for both instruments.

## 4. Discussion (Analysis of TOC Variability)

### 4.1. Comparison of Monthly Averaged Satellite Data

The most accurate and detailed estimation of the quality of measurements can only be provided by comparing single measurement results. At the same time, the averaged results on various spatial- and temporal-scale measurements can be used in the analysis of global processes and climate change. Below, we consider the averaged results of IKFS-2 measurements together with the results of independent satellite measurements.

### 4.1.1. IKFS-2 and OMI

In winter 2019–2020, the attention of the global scientific community was drawn to the low values of stratospheric ozone content in the Northern Hemisphere. In Figure 7, monthly averaged IKFS-2 and OMI TOCs around the North Pole are shown for three winter months in 2019–2020. Note that both IKFS-2 and OMI data are of Level 3, i.e., they are averaged at a regular spatial grid with a resolution of 1°. In the case of OMI, the Level 3 TOMS-Like Total Column Ozone gridded daily averaged product OMTO3d is used; it is available at https://acd-ext.gsfc.nasa.gov/anonftp/toms/omi/data/ozone/ (accessed on 1 December 2022). These data can be easily averaged over a period of consideration (in our case, over several months in 2019–2020). We averaged IKFS-2 TOC retrievals monthly on the OMI spatial grid.

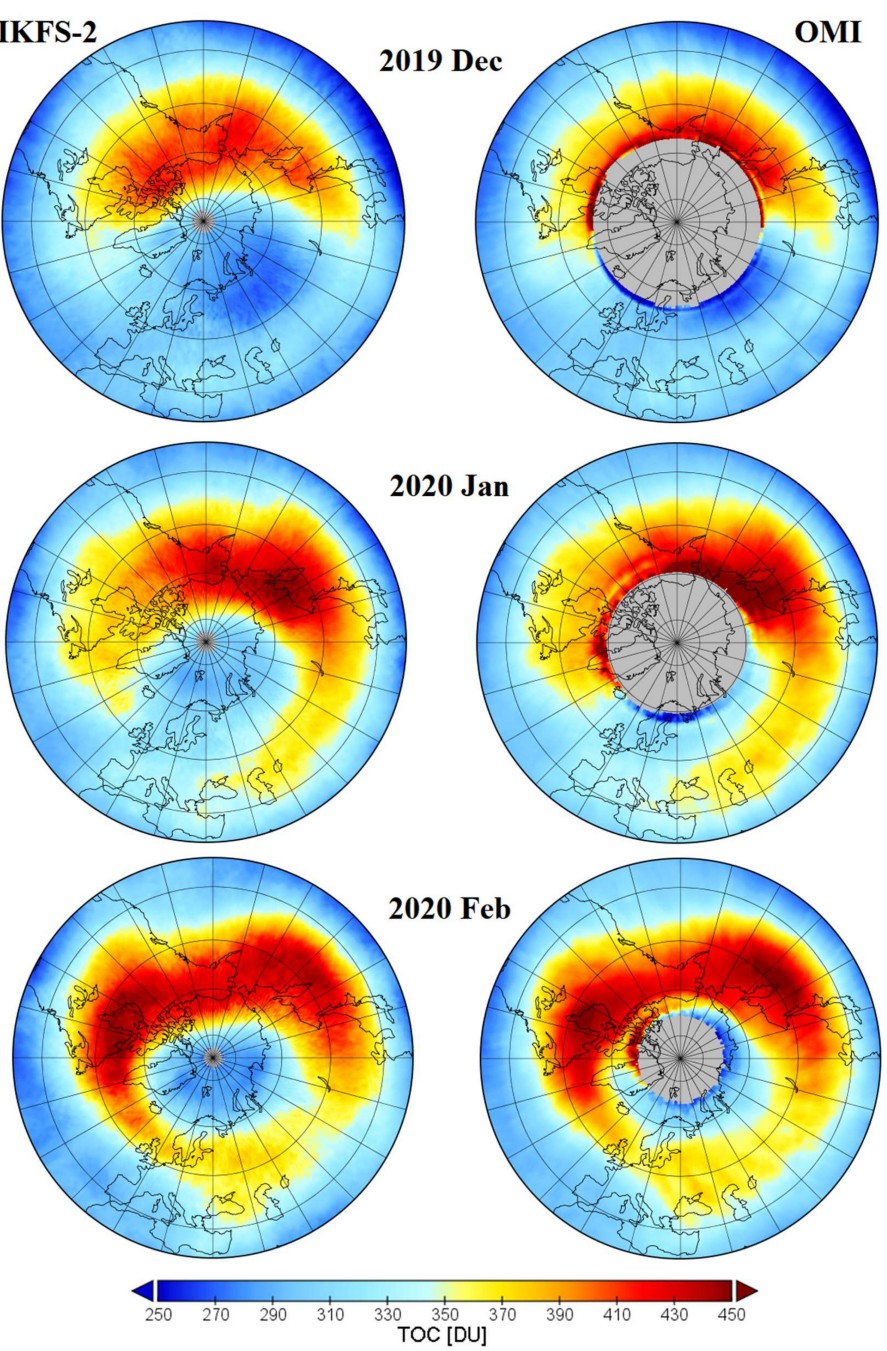

**Figure 7.** Spatio-temporal distribution of TOCs over the North Pole in winter 2019–2020 derived by the IKFS-2 and OMI.

The OMI data demonstrate low TOC values (<250 DU) over the Arctic part of Russia and Atlantic Ocean. In addition, the high levels of TOCs (>450 DU) can be seen over almost the whole other part of the territory around the North Pole. However, the data over the North Pole are absent. By contrast, the IKFS-2 data do not have such low and high TOC values around the North Pole, but they cover the whole North Pole area. The low and high TOCs near the North Pole observed from the OMI data are most likely due to the small number of and large errors in the Level 2 measurements in this region and period (a polar night).

The monthly averaged TOCs near the South Pole for each October of 2015–2020 are shown in Figure A1. October is a month with maximum depth of ozone hole formation in the Southern Hemisphere. Figure A1 depicts that the TOC distribution over the South Pole derived by IKFS-2 is close to that of the OMI data.

### 4.1.2. IKFS-2 and IASI

Another global TOC dataset is based on IASI observations from the EUMETSAT MetOp-A, MetOp-B, and MetOp-C satellites (https://www.eumetsat.int/iasi-instrument-status-calibration, accessed on 1 December 2022). However, the MetOp-C was launched in 2018; thus, its observations do not cover the whole period of IKFS-2 data, in contrast to MetOp-A (since 2006) and MetOp-B (since 2012). The IASI interferometer measures outgoing Earth thermal (IR) radiation in the 645–2760 $cm^{-1}$ range with a 0.25 $cm^{-1}$ spectral resolution (https://space.oscar.wmo.int/instruments/view/iasi, accessed on 1 December 2022).

All three MetOp satellites operate at approximately the same altitude above the Earth's surface (817–827 km) with a sun-synchronous orbit and an equatorial crossing time of 07:50 desc of local time by MetOp-A and 9:31 desc by MetOp-B and MetOp-C. IASI observations cover Earth globally twice per day with a 12 km spatial resolution. The software FORLI-O3 is used to retrieve TOCs (in a layer from the surface of up to 40 km) (Fast Optimal Retrievals on Layers for IASI O3) [37]. In [34], a good agreement between global TOCs retrieved from MetOp-A and MetOp-B IASI data was found for 2013–2017. The mean difference between the data constituted 0.4 DU, with larger differences found in the polar regions (up to 2 DU). The authors suggest that the increase in the data misfits near the poles could have been caused by difference in time passage between MetOp-A and MetOp-B over the same areas of Earth.

In this study, Level 3 monthly averaged TOC data with a 1° spatial resolution retrieved using IASI observations and FORLI-O3 v20151001 software were used (https://iasi.aeris-data.fr, accessed on 1 August 2022). The analysis of differences between the MetOp-A and MetOp-B TOC Level 3 data for 2015–2020 (Figure A2) demonstrates that the datasets correspond well to each other. Global mean differences constitute 0.02–0.03%, with increasing discrepancies near the polar regions (up to 0.6%, in particular in the Southern Hemisphere). Despite this, the MetOp-A data are almost completely unavailable for 2015. Therefore, in this study, TOCs based on MetOp-B observations were used for the comparison with the IKFS-2 dataset.

To compare IKFS-2 and IASI data, they must be coincident in time and space. For that purpose, original IKFS-2 TOC retrievals were averaged monthly on the regular IASI spatial grid (1°). In addition, the IKFS-2 data were split by solar zenith angle on the nighttime ($\geq 90°$) and daytime (<90°) datasets as the IASI data were.

Figure 8 depicts the zonal and temporal distribution of relative bias (in %) between monthly averaged TOC values derived from IKFS-2 and MetOp-B IASI for 2015–2020 separately for daytime (a) and nighttime (b) observations. The smallest differences are found near the Equator and in the region between 40°N–40°S. These differences vary from 0 to 4%. The maximal misfits are found near the poles; they reach approximately 16% or more. Moreover, the biases are larger in the Southern Hemisphere. As can be seen in Figure 9, the differences depend on the season, in particular in the polar regions.

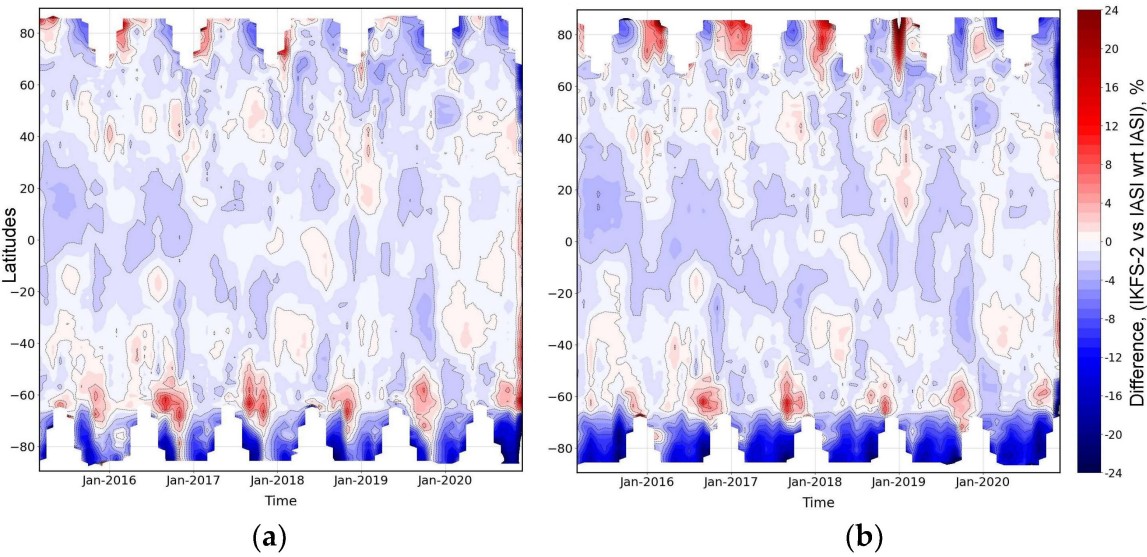

**Figure 8.** Zonal and temporal distribution of relative differences between monthly averaged TOCs derived by IKFS-2 and IASI, divided into daytime (**a**) and nighttime (**b**) observations for 2015–2020; the differences are given relative to IASI data.

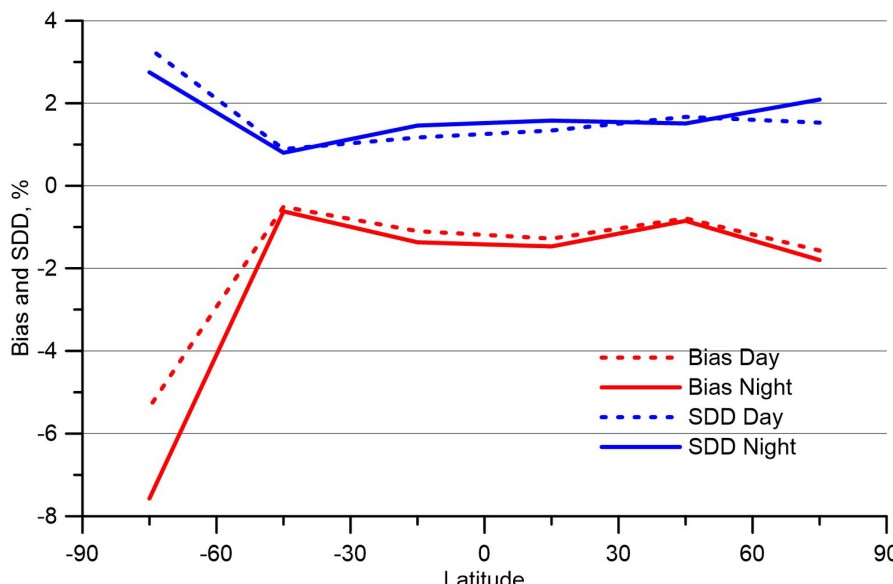

**Figure 9.** Zonal distribution of bias and SDD between daytime and nighttime IKFS-2 and IASI data for 2015–2020; the differences are given relative to the IASI data.

Table 5 and Figure 9 demonstrate the zonal distribution of the mean biases and SDDs between TOCs from the daytime and nighttime IKFS-2 and MetOp-B IASI data over 30° latitudinal bands for 2015–2020. On average, IKFS-2 data underestimate global TOC relative to IASI by 0.1–1.1% with SDD 1.65–1.68%.

The best agreement between the IKFS-2 and IASI data is found in the 30°–60° regions of both hemispheres, with biases varying from 0.5% to 0.9% and SDDs varying from 0.8% to 1.7%. The largest biases and SDDs are found in the Southern Hemisphere. The maximal biases, as was mentioned above, are found near the poles; they reach 1.6–1.8% and 5.4–7.6% in the Northern and Southern Hemispheres, respectively. The SDDs in the polar regions constitute 1.5–2.1% and 2.8–3.3% in the Northern and Southern Hemispheres, respectively. The biases between the daytime observations are slightly smaller than between the nighttime data (on average by 0.16%).

**Table 5.** Zonal distribution of bias and SDD between daytime and nighttime TOC measured by IKFS-2 and MetOp-B IASI for 2015–2020; the differences are given relative to the IASI data.

| | Day | | Night | |
|---|---|---|---|---|
| **Area** | **Bias, %** | **SDD, %** | **Bias, %** | **SDD, %** |
| 90–60°N | −1.57 | 1.53 | −1.80 | 2.09 |
| 60–30°N | −0.79 | 1.67 | −0.85 | 1.51 |
| 30–0°N | −1.28 | 1.34 | −1.47 | 1.58 |
| 0–30°S | −1.10 | 1.17 | −1.37 | 1.46 |
| 30–60°S | −0.51 | 0.89 | −0.62 | 0.80 |
| 60–90°S | −5.36 | 3.33 | −7.57 | 2.75 |
| 90N–90°S | −0.92 | 1.65 | −1.08 | 1.68 |

Comparing IKFS-2 and IASI data by seasons (Figure 10), the best agreement between datasets is observed in autumn, with biases of 0.76–0.96% (Figure 10a,b) and SDDs of 1.47–1.53% (Figure 10c,d). The worst agreement between datasets is observed in spring, with biases of 1.0–1.2% and SDDs of 1.5–1.6%. In polar regions (>60°), the largest biases (4–8%) and SDDs (5–7%) are observed, to a large extent in the Southern Hemisphere during all seasons. The biases and SDDs in the 60°N–60°S latitudinal range vary insignificantly during all seasons (from 1% to 2%). As shown in Figure 10, the correspondence between daytime IKFS-2 and IASI data is better (left) than that between nighttime data (right).

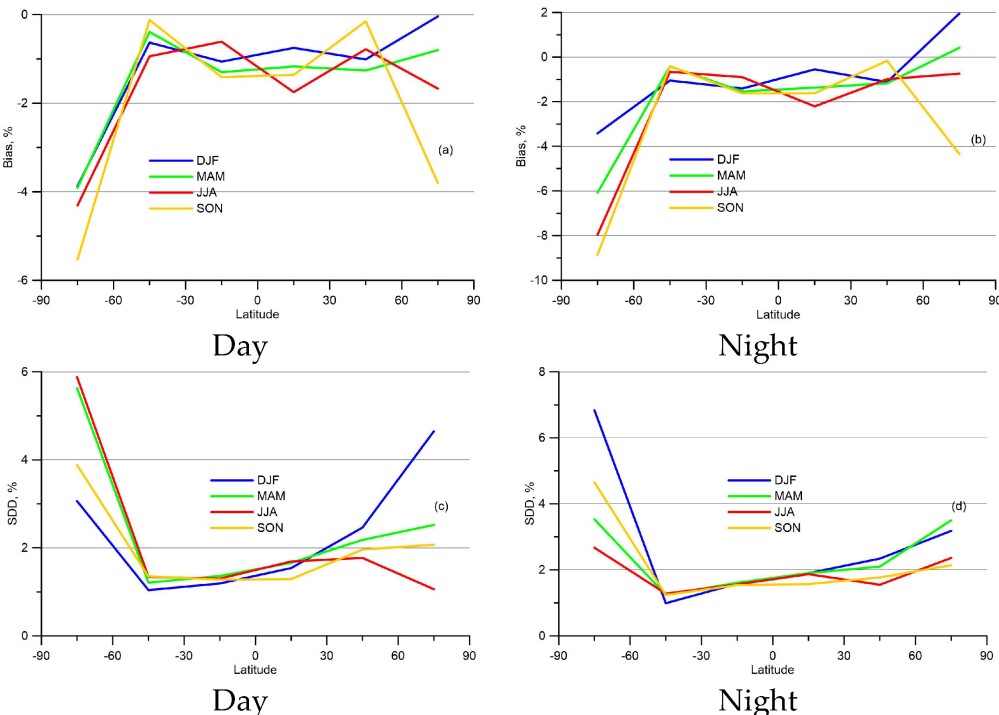

**Figure 10.** Zonal and seasonal distribution of relative differences between daytime (**left**) and nighttime (**right**) IKFS-2 and IASI data for 2015–2020; bias (**a,b**), SDD—(**c,d**); the differences are given relative to IASI data.

Figures A3 and A4 illustrate that in general, the spatio-temporal TOC distribution of the IKFS-2 and MetOp-B IASI data correspond well in all seasons for daytime and nighttime observations. Larger differences are found in the polar regions and in some areas of Africa and Eurasia, where IKFS-2 TOCs are usually overestimated in the IASI data. These areas correspond to the locations of deserts. In studies [15,34], a similar phenomenon was found from the comparison of global TOC distribution by IASI and GOME-2A satellite data.

The best agreement is found in the daytime data, with the exception of spring (Figures A3 and A4b), when maximum differences are found near the South Pole for the daytime dataset (14–18%). As we discussed above in Section 3.2.1, a high-altitude snowy surface and a cold troposphere cause a decrease in IR measurement informativity. Although both instruments measure outgoing thermal radiation, a low informativity of the spectral measurements yields an increase in contribution of a priori information and possibly cause an increase in differences between datasets.

### 4.2. Analysis of IKFS-2 TOC Retrievals

Unlike the most common satellite devices for ozone monitoring (OMI, GOME-2, ACE-FTS, etc.), which use solar radiation measurements, IKFS-2 measures thermal radiation, thus providing information on the global distribution of atmospheric ozone in periods of polar nights as well. Despite the regional and global spatial distributions of TOCs derived by IKFS-2, which we have already demonstrated in this study, in this section, the most interesting period for ozone monitoring in high latitudes—early spring—will be analyzed in more details.

Figure 11 depicts the evolution of an ozone hole measured by IKFS-2 over Antarctica in 2015–2020. IKFS-2 TOCs were averaged over 1 month from 15 September to 14 October of each year and presented on a $1° \times 1°$ grid for the 50°S–90°S latitudes. The maximum depth and width of an ozone hole are usually observed at the end of September and at the beginning of October; thus, monthly averaged ozone distribution reflects the year-to-year differences in ozone deficit over Antarctica.

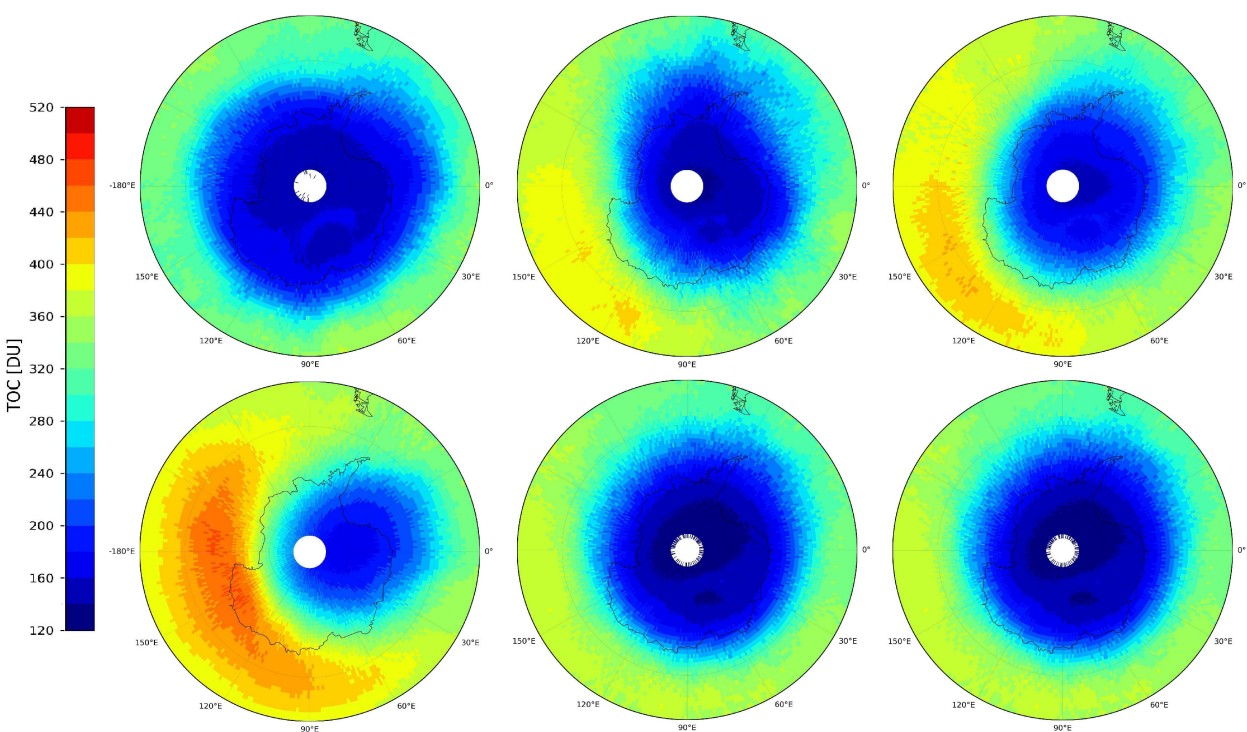

**Figure 11.** Monthly averaged TOCs derived by IKFS-2 over the Antarctic in 15 September–14 October 2015–2020 (from **left** to **right**).

Klecociuk et al. [38] calculated various metrics for estimating ozone hole strength in different years (for the 1979–2020 period). According to Metric 1 (maximum 15-day average area of ozone hole), the year 2015 occupies the 3rd, 2018—14th, 2020—15th, 2016—23rd, 2017—32nd, and 2019—the 36th place among all years considered. Figure 12 demonstrates the same picture, highlighting anomalies in TOCs considered for different latitudes. TOC values for each year are related to averaged TOCs over the whole period of IKFS-2 measurements (2015–2020). For high latitudes (70°S–90°S), the largest negative anomalies (−18%)

are observed in 2020; for the 50°S–90°S and 50°S–70°S latitudes, the largest negative anomalies (−15%) are found in 2015. Figures 11 and 12 demonstrate the capability of IKFS-2 to study year-to-year ozone anomalies during Antarctic springs.

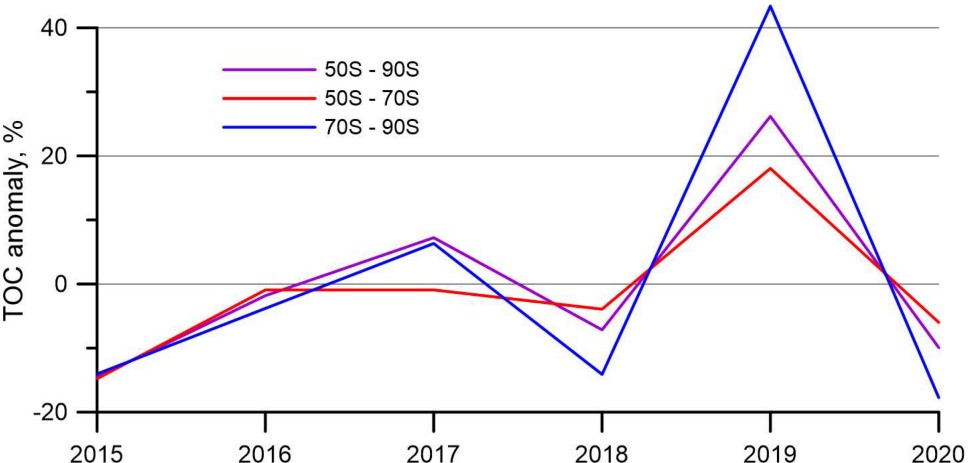

**Figure 12.** Anomalies in TOCs measured over specific latitudes in Southern Hemisphere averaged for 15 September–14 October 2015–2020.

In recent decades, ozone anomalies in spring have also been observed in the Arctic and adjacent regions. These anomalies are less pronounced and more variable than those in the Antarctic region, but they also contribute to the increase in UV radiation at the surface [4,39] and consequently to the increase in UV index even in midlatitudes [40]. Opposite to the Antarctic, the dynamics in the stratosphere during Arctic winters are some of the major factors that influence the ozone loss in the Arctic springs [41]. Polyakov et al. [31] compared the cold Arctic winter of 2019/2020 and the warm Arctic winter of 2018/2019 and, based on IKFS-2 TOC measurements, showed the deficit in TOCs in 2020 versus 2019 for the 50°N-90°N latitudes. It totaled 17%, 16%, 20%, 17% for December, January, March, and April, respectively. Even though there are no satellite observations at a distance of ~400 km around the poles due to the Meteor M N2 satellite's geometry of flight (see Section 2.1), for simplicity, intervals of latitudes up to 90° north and south latitudes are correspondingly indicated in Figure 12.

Figure 13 depicts the variability in ozone distribution over the mid and high latitudes (50°N–90°N) of the Northern Hemisphere in March. Compared to March 2020, with significant ozone loss observed in spring in the Northern Hemisphere [5,42], ozone anomalies in other years are not as evident, but they can influence the level of UV irradiance at the surface in different areas. In 2015, ozone anomalies in March were observed in the European part of Russia (30°E–60°E); in 2016 and 2017, to a lesser degree, they spread to Northern and Central Europe and Western and Central Siberia (20°W–110°E). In 2018, ozone anomalies concentrated over the northern part of Central Siberia; in 2019, they shifted to Central Europe and North America (90°W–150°W), and finally, in 2020, essential ozone loss was observed mostly over the northern part of the whole Northern Hemisphere.

Table 6 presents monthly TOCs for March of each year averaged over 50°N–70°N at specific longitudes, which are related to different geographical regions. With the exception of East Siberia, other regions are densely populated, especially North, Central and Eastern Europe and the European part of Russia, where 17 cities with population of more than 1 million inhabitants in each are located. The largest ozone anomalies in each year (300–315 DU) were observed over the less populated region of the far east of Russia and East Siberia. The territory of Canada suffered severely from ozone loss in March 2020 (340 DU). Lower than usual TOCs over the European regions in March were observed in 2016 and to a lesser extent in 2015 and 2016 (360–370 DU) over Central and West Siberia in 2016 and 2017 (350–360 DU). Bernhard et al. [4] estimated the increase in UVI (UV index)

due to ozone loss in the spring of 2020 and showed that UVI anomalies in northern Canada and the central part of Russia in March exceed 40–70% relative to the 2005–2019 averages. Chubarova et al. [39] calculated UVI near local noon in March 2016 for several ground-based stations in Russia located between the 60°N and 70°N latitudes and between the 55°E and 100°E longitudes (mainly West and Central Siberia). They showed that even for such high latitudes and for the middle of March, the UVI reached 2.9, which exceeded the threshold for skin type 1 (2.2) and was close to erythema threshold for the most common skin type 2 (3.0). Another example of UVI in the period and regions considered are the measurements of UVI at the Diekirch GAW European station in Luxembourg (49.9°N and 6.2°E) taken from the WOUDC site. At the end of March 2015–2017, in separate days, the UVI exceeded 3—the erythemal threshold for skin type 2. All these examples demonstrate that even in March and in mid and high latitudes when the Sun elevation is relatively low, UV irradiance can damage human skin if monthly means total 350–370 DU.

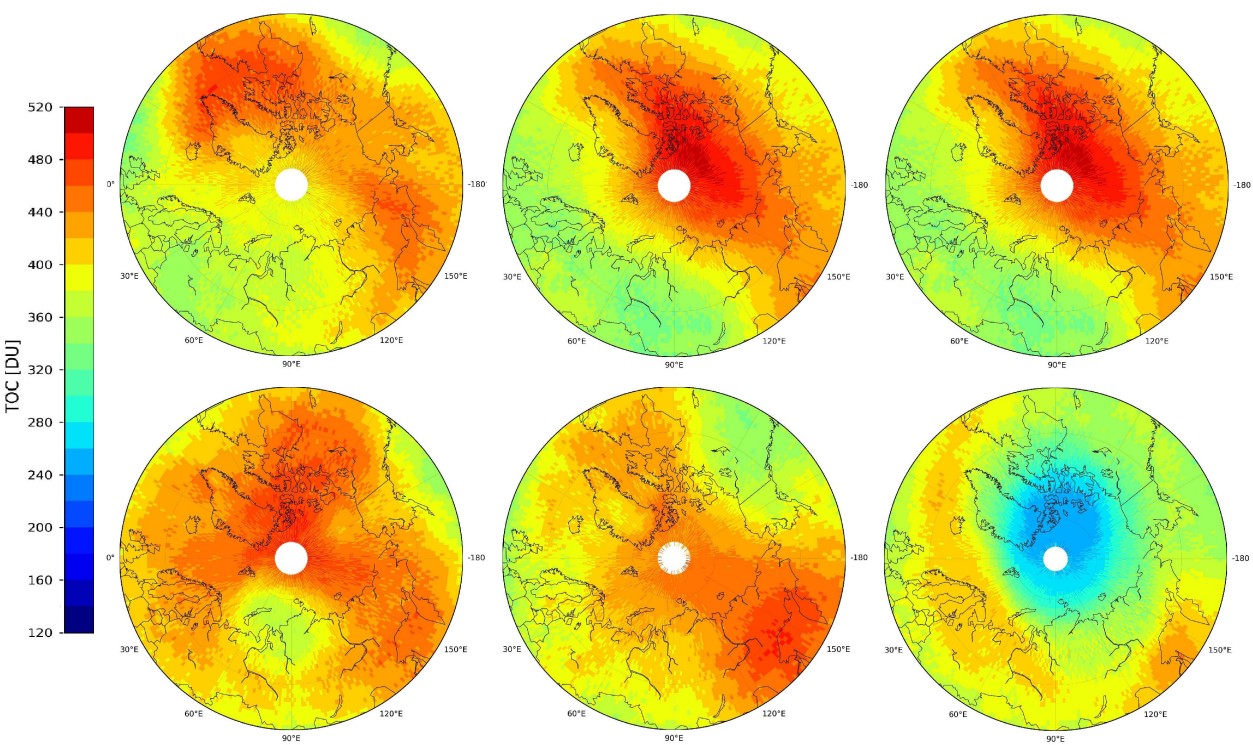

**Figure 13.** Monthly averaged TOCs derived by IKFS-2 over the Arctic in Mar 2015–2020 (from **left** to **right**).

**Table 6.** Averaged TOC values (in DU) in Mar 2015–2020 observed over 50°N–70°N and different geographical regions.

| | Latitude Range (Region) | | | | |
|---|---|---|---|---|---|
| **Year** | **130°W–60°W (Canada)** | **10°W–30°E (Northern and Central Europe)** | **30°E–60°E (European Part of Russia)** | **60°E–120°E (Western and Central Siberia)** | **120°E–170°W (Eastern Siberia and Far East)** |
| 2015 | 434.3 | 376.3 | 361.5 | 380.9 | 315.4 |
| 2016 | 429.6 | 358.9 | 358.3 | 354.6 | 314.7 |
| 2017 | 416.0 | 372.5 | 359.4 | 362.9 | 308.8 |
| 2018 | 440.0 | 429.4 | 416.6 | 399.6 | 314.0 |
| 2019 | 394.9 | 385.8 | 398.2 | 406.6 | 318.0 |
| 2020 | 340.8 | 396.6 | 399.0 | 388.8 | 298.9 |

## 5. Conclusions

A technique for total ozone column (TOC) retrieval from outgoing thermal Earth radiation spectra that was previously proposed was improved, optimized, and applied to the 2015–2020 period of measurements. The technique is based on the ANN algorithm and the method of principal components. It uses spectral measurements of the IKFS-2 Fourier-spectrometer on board the Meteor M N2 weather satellite. Using OMI TOCs for the ANN training solves the issue of IKFS-2 retrieval calibration. Approximation errors equaled 8.36 DU or 2.46%. The technique presented can be adapted and applied to the spectral measurements of other satellite instruments with similar characteristics, such as IASI, AIRS, etc.

The global IKFS-2 TOCs in 2015–2020 were estimated and validated against independent ground-based and satellite measurements. The coincidence criteria for the IKFS-2 and hourly Dobson and Brewer ground-based direct sun measurements were set to 70 km and 1 h. The mean bias for the data pairs constituted −0.23% with an SDD of 2.9% for the WOUDC network. The mean bias and SDD between the IKFS-2 TOCs and the Eubrewnet network ground-based measurements were −0.40% and 2.7%, respectively.

The comparison of the IKFS-2 TOCs to the daily averaged Dobson and Brewer data with a spatial mismatch of 150 km shows a mean bias of −0.59% with an SDD of 4.5%. The co-location criteria for the IKFS-2 and TROPOMI (Sentinel-5P satellite) data pairs were 18 km and 6 h for space and time, respectively. The mean bias between data sets for 2018–2020 equals −1.2% with an SDD of 3.1%.

The analysis of the differences between IKFS-2 and both daily ground-based and Tropomi TOC observations allows us to estimate the preliminary latitudinal and seasonal behavior of the IKFS-2 TOCs accuracy and precision. It is shown that the mean bias generally does not exceed 2%, with SDDs usually less than 2% in the tropics. SDDs increase towards the poles, especially in the autumn and winter seasons. The maximal SDDs reach 4–6% in the Arctic and 6–8% in the Antarctic region. Note that these estimates are obtained based on the comparison of the IKFS-2 TOCs to the data that also contain errors.

Monthly averaged TOCs near the South Pole derived from IKFS-2 and OMI are in good agreement during the seasonal ozone depletion in October. Unlike OMI, under specific conditions, such as polar nights, IKFS-2 is able to provide TOC data without solar radiation.

In general, monthly averaged global TOCs derived from IKFS-2 are in good agreement with IASI MetOp-B day and nighttime data for the whole period of IKFS-2 measurements. On a global scale, IKFS-2 underestimates TOC with respect to IASI data by 0.9–1.1% with SDDs of ~1.7%. The best agreement is found in the tropics and the middle latitudes, with differences increasing towards the poles (the largest in the Southern Hemisphere). The smallest discrepancies between the IKFS-2 and IASI datasets on a global scale are found in autumn, and the largest misfits are observed in spring. Finally, the analysis revealed that daytime TOCs derived from two instruments agree slightly more so than those during nighttime (by ~0.16%).

This study has demonstrated that by using IKFS-2 data, it is possible to monitor relatively rare ozone depletion events in the Northern Hemisphere, and it is possible to analyze the detrimental effect that such phenomena may have on the inhabitants of the Boreal regions of the Earth.

The comparisons of IKFS-2 TOCs with the data from different measurement techniques demonstrate the good ability of the IKFS-2 measurements and the ANN algorithm to depict the global distribution of TOCs in all seasons of the year.

In addition, the analysis of the ozone anomalies in both hemispheres detected by the IKFS-2 measurements are given.

**Supplementary Materials:** The following supporting information can be downloaded at: https://www.mdpi.com/article/10.3390/rs15092481/s1, Video S1. Ozone Total Column around the North Pole during March 2020.

**Author Contributions:** Conceptualization, A.P., Y.V., D.K. and Y.T.; data curation, A.P. and D.K.; funding acquisition, Y.T.; investigation, A.P.; methodology, A.P.; project administration, A.P.; software, A.P. and Y.V.; validation, A.P. and Y.V.; visualization, A.P., Y.V. and G.N.; writing—original draft, A.P., Y.V. and G.N.; writing—review and editing, A.P., Y.V. and G.N. All authors have read and agreed to the published version of the manuscript.

**Funding:** The study was performed at the 'Laboratory for the Research of the Ozone Layer and the Upper Atmosphere' of Saint Petersburg State University and was supported by the Government of the Russian Federation under agreement [075-15-2021-583].

**Data Availability Statement:** IKFS-2 spectral measurements are available through the open-access website of the SRC 'Planeta' http://planet.rssi.ru/calval/public-ikfs, last access: [23 November 2022]. The IKFS-2 ozone total column data and coefficients binary file (see Appendix B) are available upon a request from the authors. Ground-based ozone measurements can be downloaded from https://woudc.org, last access: [11 February 2022] (WOUDC data) and http://eubrewnet.aemet.es/, last access [3 March 2022] (Eubrewnet data). OMI ozone measurements can be found athttps://doi.org/10.5067/Aura/OMI/DATA2024 [3 December 2020] and TROPOMI ozone total column measurements are available at https://tropomi.gesdisc.eosdis.nasa.gov/data/S5P_TROPOMI_Level 2/, last access [17 July 2022]; IASI MetOp-B Level 3 ozone total columns measurements are available at https://iasi.aeris-data.fr/catalog, last access: [1 August 2022].

**Acknowledgments:** IKFS-2 spectral measurements were provided by the SRC "Planeta" through the open-access website. SRC "Planeta" acts as an operator of the IKFS-2 instrument and other onboard hydrometeorological and oceanographical equipment that is mounted onto meteorological and environmental satellites, including Meteor-M N2. Ground-based ozone measurements were provided by the WMO/GAW Ozone Monitoring Community, World Ozone and the Ultraviolet Radiation Data Centre (WOUDC); a list of all contributors is available on the website. doi:10.14287/10000001. We thank the European Brewer Network (http://eubrewnet.aemet.es/, accessed on 3 March 2022) for providing access to the data and the PI investigators and their staff for establishing and maintaining the 29 sites used in this investigation. OMI and TROPOMI ozone measurements were provided by Goddard Earth Sciences Data and Information Services Center (GES DISC). IASI MetOp-B ozone data are provided by the scientific community of The Laboratoire atmosphères, milieux, observations spatiales (LATMOS), a list of contributors is available at https://iasi.aeris-data.fr/catalog/#aeris-metadata-contacts, accessed on 1 August 2022.

**Conflicts of Interest:** The authors declare no conflict of interest.

**Appendix A**

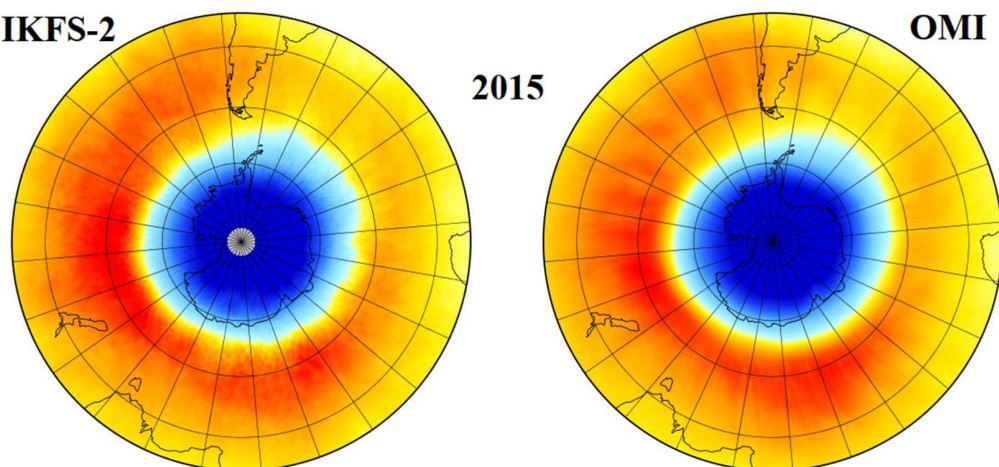

**Figure A1.** *Cont.*

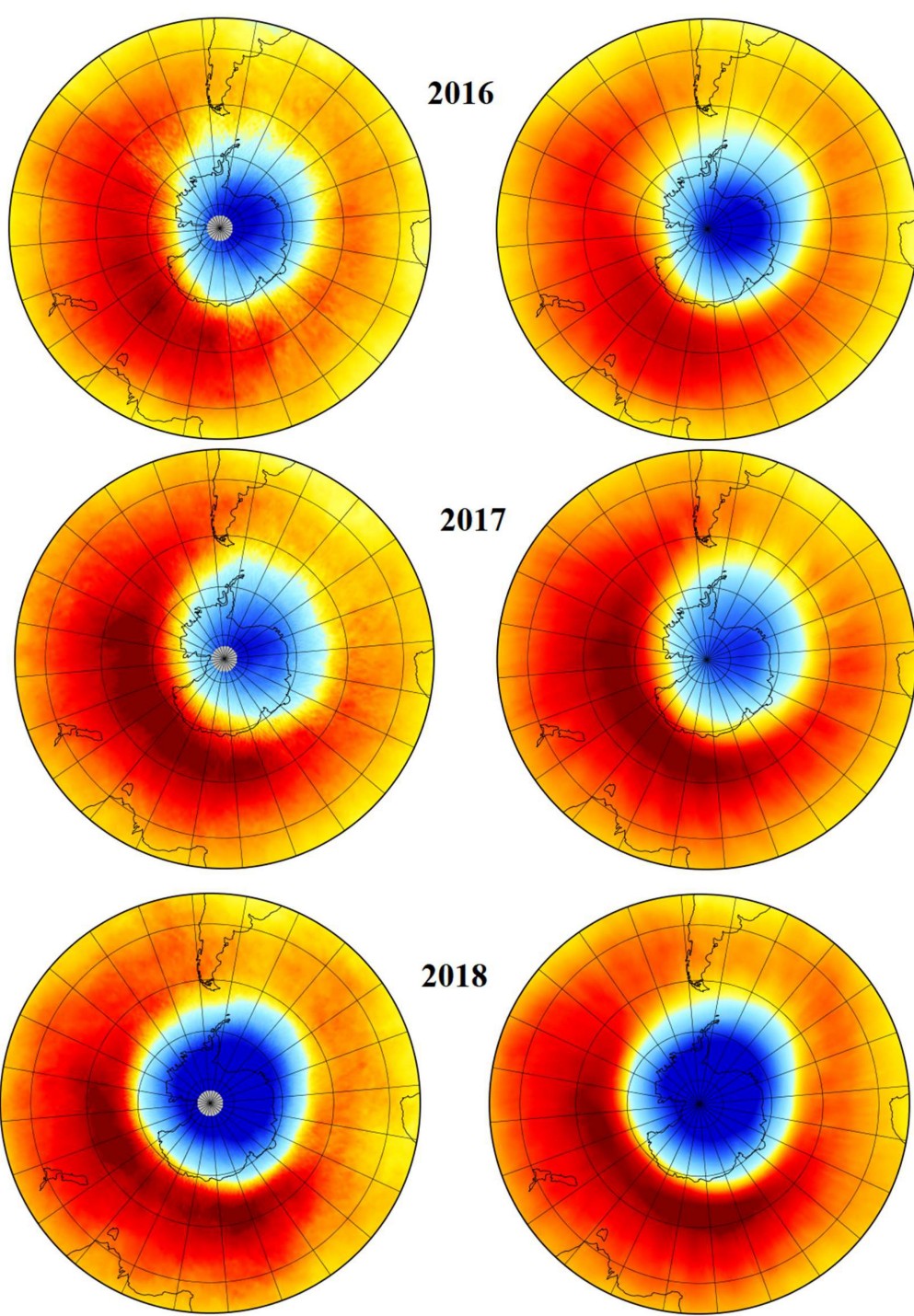

**Figure A1.** *Cont.*

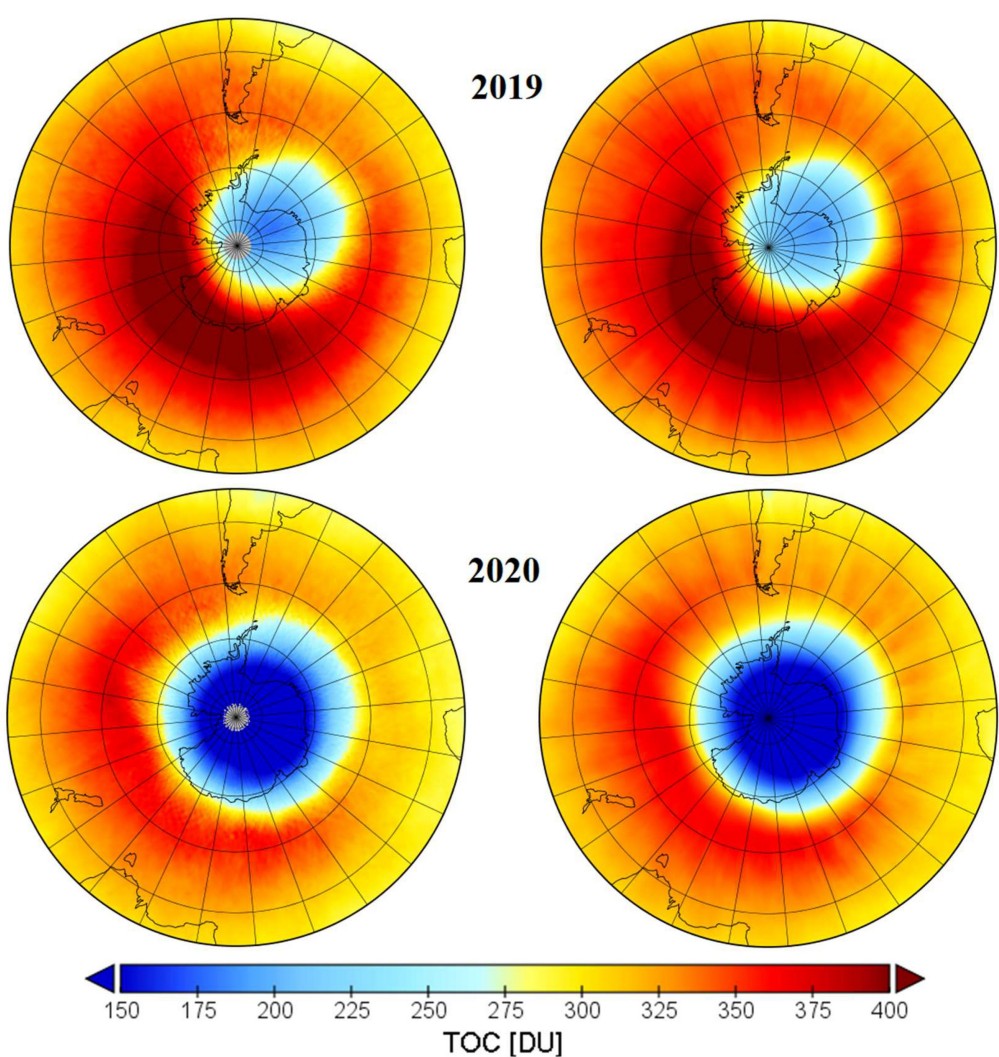

**Figure A1.** Spatio-temporal distribution of TOC over the South Pole in October 2015–2020 derived by IKFS-2 and OMI.

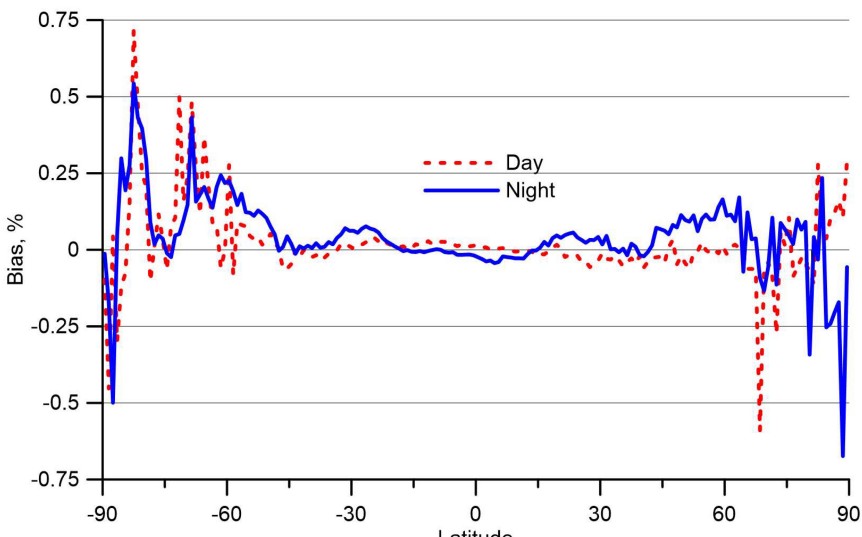

**Figure A2.** Zonal distribution of mean differences between TOCs by MetOp-A and MetOp-B measurements for 2015–2020.

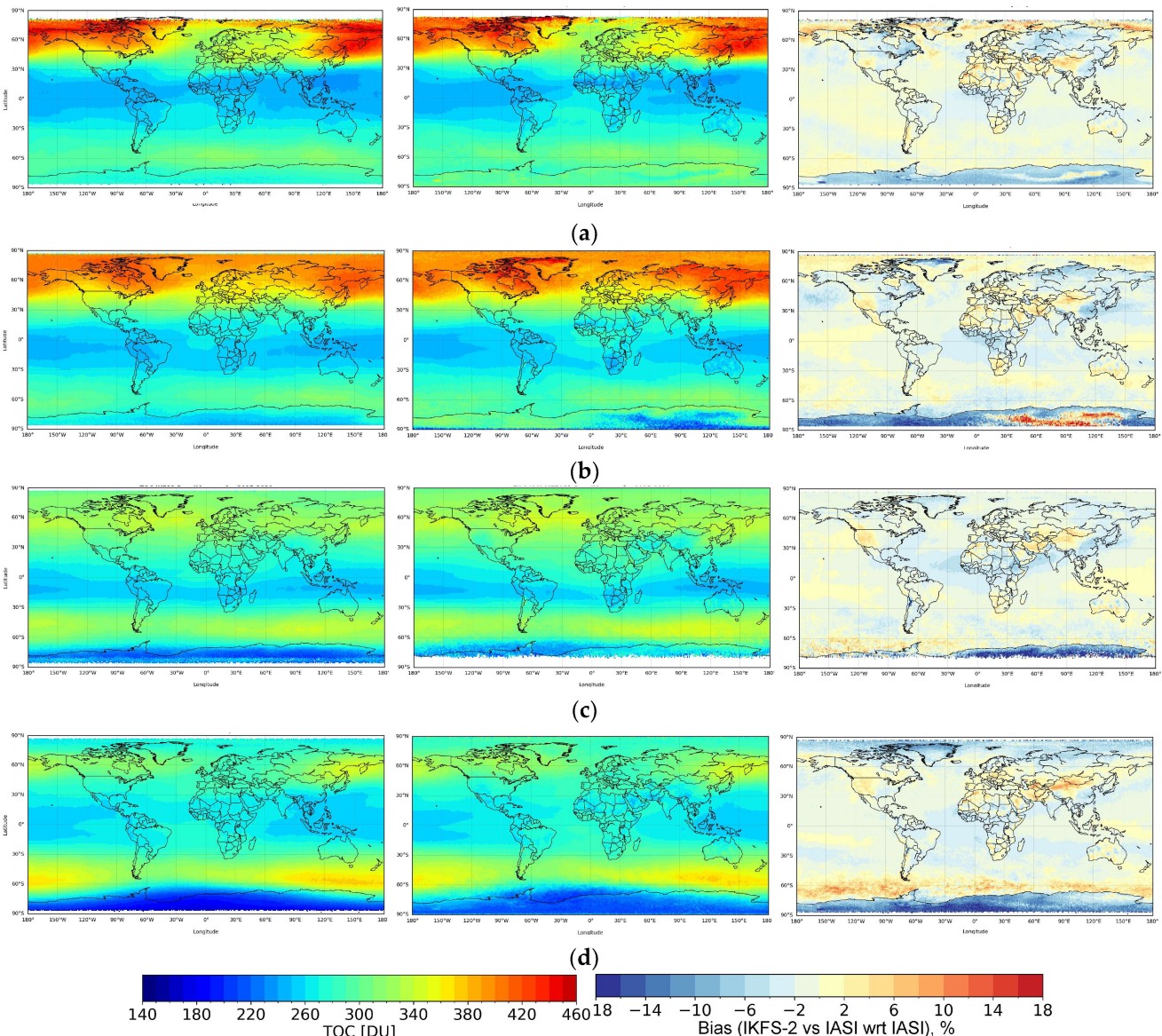

**Figure A3.** Global TOC distribution by seasons ((**a**)—DJF, (**b**)—MAM, (**c**)—JJA, (**d**)—SON) for 2015–2020 by daytime IKFS-2 (**left**) and IASI (**middle**) data as well as their difference in % relative to the IASI data (**right**).

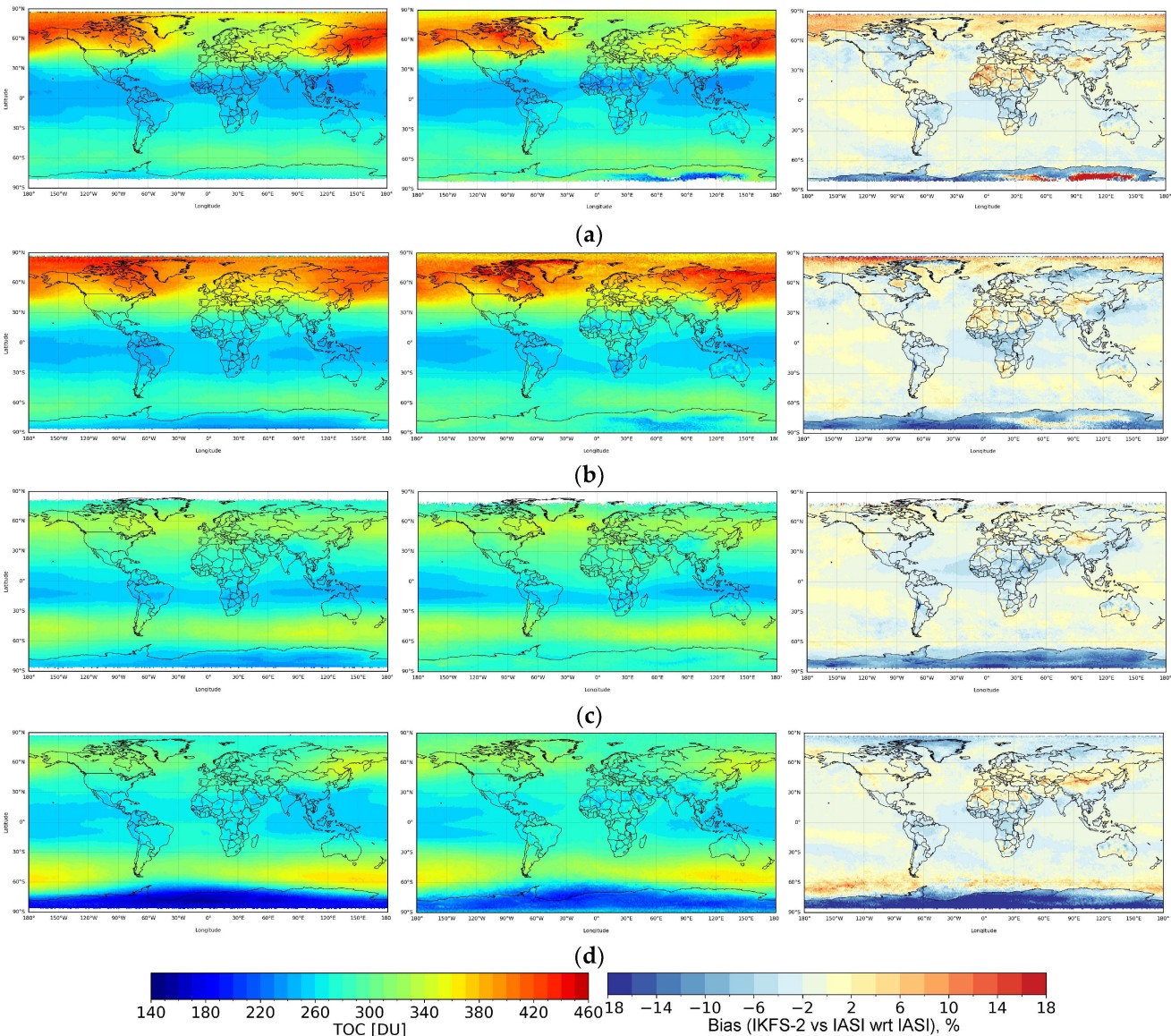

**Figure A4.** Global TOC distribution by seasons ((**a**)—DJF, (**b**)—MAM, (**c**)—JJA, (**d**)—SON) for 2015–2020 by nighttimeIKFS-2 (**left**) and MetOp-B IASI (**middle**) data as well as relative difference in % relative to the IASI data (**right**).

## Appendix B

*Appendix B.1. Retrieval Algorithm Detailed Description*

Appendix B.1.1. Input Parameters

The zenith angle $Za$ of the satellite, which is derived from the observed pixel of the Earth's surface, the pixel latitude, and the fraction of year are used. The first set consists of $N_{PC_{total}}$ PCs in the 600–1210 cm$^{-1}$ spectral region. This region includes the $CO_2$ spectral absorption band that contains information on the atmospheric temperature profile, the transparency window with information on the surface, and the 9.6 μm ozone absorption spectral band. The second set consists of $N_{PC_{O3}}$ PCs in the 980–1080 cm$^{-1}$ spectral region of the ozone absorption band (see Section 2).

Appendix B.1.2. Algorithm Step by Step

Step 1 of the retrieval algorithm is the calculation of PCs. Empirical orthogonal functions ($EOF_{i,k}$) and mean spectrum $\bar{J}_k$ are used to calculate PC values:

$$PC_i = \sum_{k=k_1}^{k_2} (J_k - \bar{J}_k) EOF_{i,k}, \tag{A1}$$

where $i$ is number of PCs and EOFs, $k$ is number of spectra points, and $k_1$ and $k_2$ are the first and last numbers of the selected spectra region. Spectral region parameters are shown in Table A1.

**Table A1.** Spectral regions and PC parameters.

| Spectral Region | Name | $N_{PC}$ | Spectral Point Numbers (Wavenumber, cm$^{-1}$) | |
|---|---|---|---|---|
| | | | First point $k_1$ | Last point $k_2$ |
| 660–1210 | Global | $N_{PC_{total}} = 25$ | 1 (660) | 1571 (1210) |
| 980–1080 | O$_3$ band | $N_{PC_{O3}} = 50$ | 915 (980) | 1200 (1080) |

Step 2 is the collection of input parameters vector X, see Table A2.

**Table A2.** Input parameters of solution operator.

| Parameter | Name | Input Vector |
|---|---|---|
| fraction of year (Day number/365 or 366) | f | $x_1$ |
| Pixel latitude, degrees | lat | $x_2$ |
| A zenith angle of satellite from pixel, degrees | Za | $x_3$ |
| PCs in the 600–1210 cm$^{-1}$ spectral region | PC$_{total}$ | $x_4, x_5, \ldots, x_{3+N_{PC_{total}}}$, |
| PCs in the 980–1080 cm$^{-1}$ spectral region | PC$_{O3}$ | $x_{4+N_{PC_{total}}}, \ldots, x_{3+N_{PC_{total}}+N_{PC_{O3}}}$, |

Step 3 is the normalization of $X$ vector to interval $(-1,1)$ by relation (A2)

$$x_i = 2\left(X_i - X_i^{min}\right)/\left(X_i^{max} - X_i^{min}\right) - 1, \tag{A2}$$

where $X_i^{max}$ and $X_i^{min}$ are maximal and minimal values of $X_i$.

Step 4 is the main calculation of ANN by relation (A3)

$$y = f\left(\sum_{j=1}^{n_h} W_j^2 (f(\sum_{i=1}^{n_x} W_{i,j}^1 x_i + b_j^1)) + b^2\right), \tag{A3}$$

where $y$ is an normalized result, $f$ is an activation function (*th*), $W$ and $b$ are coefficients, $W^1$ and $b^1$ mean first layer, $W^2$ and $b^2$ mean second layer.

Step 5 is the denormalization of the result, as is shown by relation (A4)

$$TOC = TOC_{min} + (y+1)(TOC_{max} - TOC_{min})/2 \tag{A4}$$

All of the pointed data, mean spectrum, and EOF are saved in a special file in binary form. Data structure is shown in Table A3.

**Table A3.** The structure of the data file for the retrieval algorithm.

| Name | Meaning | Type (Fortran) |
|:---:|:---:|:---:|
| $f$ | Activation function | Character *4 |
| $n_1$ | Layers number, always 2 | Integer *4 |
| $n_x$ | Length of vector X | Integer *4 |
| $n_h$ | Number of hidden level neurons | Integer *4 |
| $n_y = 1$ | Results number | Integer *4 |
| $n_z = 1$ | Reserved | Integer *4 |
| $X^{min}$ | Xmin | Real *4 |
| $X^{max}$ | Xmax | Real *4 |
| $Y^{min}$ | Ymin | Real *4 |
| $Y^{max}$ | Ymax | Real *4 |
| $b^2$ | ANN coefficien | Real *8 |
| $W^2$ | ANN coefficiens | Real *8 |
| $b^1$ | ANN coefficiens | Real *8 |
| $W^1$ | ANN coefficiens | Real *8 |
| | *EOF total* | |
| $n_v$ | Number of spectral points = 1571 | Integer *4 |
| $N_{PC}$ | Number of EOF = 25 | Integer *4 |
| $\bar{J}$, EOF | Mean spectra and EOF | Real *8 |
| | *EOF O3* | |
| $Nv$ | Number of spectral points = 286 | Integer *4 |
| $N_{PC}$ | Number of EOF = 50 | Integer *4 |
| $\bar{J}$, EOF | Mean spectra and EOF | Real *8 |

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
