# Peer review of "Six Years of IKFS-2 Global Ozone Total Column Measurements"

_remotesensing, doi:10.3390/rs15092481_

Round 1

Reviewer 1 Report

The reviewed manuscript discusses the use of satellite methods to monitor atmospheric ozone, which plays an important role in protecting the biosphere and contributing to the Earth's climate. The study focuses on the IKFS-2 spectrometer aboard the Meteor M N2 satellite, which uses Fourier-transform infrared (FTIR) measurements of thermal radiation to determine total ozone columns (TOCs). The study is based on an artificial neural network trained using the TOCs retrieved from the Aura OMI instrument as a reference. The results of the study show that the TOCs predicted with the help of the ANN described in this work agree well with other measurements.

Overall, this is a good methodological work, which is well written, and I wouldn’t hesitate to recommend its publishing in the journal given that the general comments below are properly addressed. I have selected “major revision”, but the changes I suggest below are easy to implement.

General comments:

1)              Even though the “Data availability statement” at the end of the manuscript provides necessary links and references, it does not contain the main result of the work, the neural network itself. However, this might be the most useful output of this activity because the researchers will be able to produce their own data from IKFS-2. Moreover, I’m almost sure that the information content of the IKFS-2 channels is close to that of the corresponding IASI or AIRS channels: despite different spectral resolution of these instruments, the kernel functions do not change that much. From this point of view, providing the converged ANN will be interesting for those working with these instruments. As far as I know, the ANN can be represented by a set of expressions that can be easily packed to a file in the supplementary (or even to the Appendix). I believe that with this change the work will (a) correspond to modern requirements to methodological manuscripts and (b) have broader impact on the remote sensing field.

2)              The authors notice that the resulting approach routinely retrieves TOCs, which are slightly underestimated. Another observed issue is that the discrepancies increase towards the poles. I understand that this is a sensitive point, but if there’s a reference dataset and the methodology, which yields the values, which are very close to reference ones, be fine-tuned through some additional recalibration procedure? Wouldn’t it be better to provide a “black box with correction” than a “black box, which is off”? (By “black box” here I mean the ANN)

3)              Section 4.1.2 discusses the differences between IKFS-2 retrievals IASI, but it does not attempt to explain them, whereas it’s the explanation, which is interesting for the reader. At the same time, up to 20% relative differences shown in Fig. 8 raise certain questions. I would add a paragraph (or even a subsection) discussing potential sources of these differences here.

Minor comments and technical corrections

Line 42: “but this trend observed” -> “but this trend”

Lines 65, 70, 76 and elsewhere: comma is missing after the introductory phrase (clause rule).

Lines 83-89: This is the first mentioning of AIRS and IASI. I would estimate the applicability of the ANN to the channels of these instruments and mention this in the text of the manuscript and in the Conclusions to advertise the method.

Lines 140 and 713: Conclusions are in Section 5.

Line 145: should be “payloads”

Line 164: as far as I understand, it’s the wavelength integration bin size that should be increased, and not the resolution, which is inverse to bin size.

Lines 203-205 : please, see the second general comment

Line 222: “There S=3 when …” – I didn’t get what was meant here. Please, rewrite.

Line 231: “Well-precise ground-based” – did you mean “Reference ground-based” or “High-precision ground-based”?

Line 263: “Both ANNs depict very close discrepancies with independent data”. I would suggest something like “Both ANNs are in close agreement with the independent data” or “The discrepancies between both ANNs and the independent data are small”.

Line 276: I guess, power of 2 is missing for the bracket in the nominator under the square root.

Lines 287-294: I don’t know the rules for using the hyperlinks in the text, but the flow is definitely broken by long links. Is it possible to use a reference in the footer?

Author Response

First of all, authors are thankful to the Referee for reading the manuscript and useful comments. Below, are the answers to comments.

General comments:

1)              Even though the “Data availability statement” at the end of the manuscript provides necessary links and references, it does not contain the main result of the work, the neural network itself. However, this might be the most useful output of this activity because the researchers will be able to produce their own data from IKFS-2. Moreover, I’m almost sure that the information content of the IKFS-2 channels is close to that of the corresponding IASI or AIRS channels: despite different spectral resolution of these instruments, the kernel functions do not change that much. From this point of view, providing the converged ANN will be interesting for those working with these instruments. As far as I know, the ANN can be represented by a set of expressions that can be easily packed to a file in the supplementary (or even to the Appendix). I believe that with this change the work will (a) correspond to modern requirements to methodological manuscripts and (b) have broader impact on the remote sensing field.

Applying the technique to a similar instrument will require a lot of calculations and processing of large amounts of data, as well as the necessary validation. In fact, it is necessary to repeat for another device all the calculations performed by us for IKFS-2, which will require at least several months of work. Nevertheless, we have included in Attachment C a detailed description of the calculation algorithm and the structure of the file containing all the necessary coefficients in binary format. A link to the specified file will also be provided.

2)              The authors notice that the resulting approach routinely retrieves TOCs, which are slightly underestimated. Another observed issue is that the discrepancies increase towards the poles. I understand that this is a sensitive point, but if there’s a reference dataset and the methodology, which yields the values, which are very close to reference ones, be fine-tuned through some additional recalibration procedure? Wouldn’t it be better to provide a “black box with correction” than a “black box, which is off”? (By “black box” here I mean the ANN)

As we understand correct, the reviewer suggested to modify the solving operator by adding the information from data that we used for validation of our retrieval method. This will certainly improve the retrieval technique, but it will require additional validation of the technique by using new data. And this iteration process of improvement and subsequent validation will continue until we use all available data. And in this case, we will get the best retrieval technique but will not be able to validate it. And this is out of the scope of the current study. We presented a new retrieval technique and validated the results of its application to measured spectra.

3)              Section 4.1.2 discusses the differences between IKFS-2 retrievals IASI, but it does not attempt to explain them, whereas it’s the explanation, which is interesting for the reader. At the same time, up to 20% relative differences shown in Fig. 8 raise certain questions. I would add a paragraph (or even a subsection) discussing potential sources of these differences here.

Done, see end of section 4.1.2

Minor comments and technical corrections

Line 42: “but this trend observed” -> “but this trend”

Done

Lines 65, 70, 76 and elsewhere: comma is missing after the introductory phrase (clause rule).

Done

Lines 83-89: This is the first mentioning of AIRS and IASI. I would estimate the applicability of the ANN to the channels of these instruments and mention this in the text of the manuscript and in the Conclusions to advertise the method.

Done, we mentioned this possibility in the Conclusions. Also see reply to the first general comment.

Lines 140 and 713: Conclusions are in Section 5.

Done

Line 145: should be “payloads”

Done

Line 164: as far as I understand, it’s the wavelength integration bin size that should be increased, and not the resolution, which is inverse to bin size.

Yes, “bin size” is called “spectral resolution”

Lines 203-205 : please, see the second general comment.

We explained our motivation in the reply to second general comment.

Line 222: “There S=3 when …” – I didn’t get what was meant here. Please, rewrite.

Done

Line 231: “Well-precise ground-based” – did you mean “Reference ground-based” or “High-precision ground-based”?

Done, we chose “high-precision”.

Line 263: “Both ANNs depict very close discrepancies with independent data”. I would suggest something like “Both ANNs are in close agreement with the independent data” or “The discrepancies between both ANNs and the independent data are small”.

Thank you, we agree. done

Line 276: I guess, power of 2 is missing for the bracket in the nominator under the square root.

Thanks, it was a misprint.

Lines 287-294: I don’t know the rules for using the hyperlinks in the text, but the flow is definitely broken by long links. Is it possible to use a reference in the footer?

Done, we remove these hyperlinks from the text as they do not relate to the current research.

Reviewer 2 Report

See the attached document. 

Author Response

First of all, authors are thankful to the Referee for reading the manuscript and useful comments. Below, are the answers to comments.

-In the title: To be more explicit for non-specialists and to emphasized that the measurements were made “all over the world” and with a satellite spectrometer, please consider the following title: Six years of ozone total column measurements all over the world made by IKFS-2 spectrometer onboard of Meteor M N2 satellite.

Done, by addition word “global”.

-In line 13: Explain what means “IKFS” in the description of the IKFS-2 spectrometer.

Done, see section 2.1

-In lines 126-128: Verify the sentence: “They showed the for direct sun ground-based measurements with 1 h temporal difference and 70 km spatial differences the mean bias totaled -1.46% with the SDD of 2.57%.”. Possibly the first part of the sentence can be written as: “They showed that for direct sun…..”.

Done

-In line 210: The legend of Figure 1 related to ANN (Artificial Neural Network) needs to include the description of each of the variables (x, f, etc).

Done

-In lines 212-213: Explain the meaning of “PCs” for non-specialists.

We expanded the existed explanation, given in the first paragraph of section 2.2

-In line 224: Change Nx by nx in the text of the legend: “…where Nh is the number of neurons of a hidden layer, see Fig. 1.” or in the figure replace nx by Nx.

Done

-In line 236: Suppress the symbol “-“ in the text: “OMI TOC data – are the data with……”

Done

-In line 310 and in line 312: Add the unit “DU” to the numbers: and 127.2 and 168.4, respectively.

done

-In line 362: Verify the text: “…we t data from 29 stations…..”

Done:  “… we use for the comparison data from 29 stations…”        

-In line 369: The name of the site: “Rio Callegos”, actually is “Río Gallegos”.

Done

- In lines 393-394: Explain in the legend of Figure 3 what means in the vertical axis “GB wrt GB”.

-In lines 456-457: The same comment as in lines 393-394 (Figure 3), for Figure 4.

-Lines 804-805: The same comment as in lines 393-394 (Figure 3), for Figure A2.

We changed all the legends.

-In line 440: Add “satellite instruments” after “other” and in place of “satellite”, in the text: “…and two other satellites - OMI and TROPOMI.”

Done

-In lines 451-452: Can you verify in your data (and consequently explain in this part of the text) if the large difference in SDD in the South Pole region are due to the presence of the Ozone Hole during winter and spring Southern Hemisphere seasons or eventually to other differences (for example, differences in the hour of measurement of TOC by the two instruments (IKFS-2 and OMI)?

Done, text was corrected.

“High-altitudes snowy surface and cold troposphere cause increasing differences in South Pole region.”

-In lines 662-663: Some of the latitude intervals (50S-90S and 70S-90S) in Figure 12 include the latitudes very near the South Pole. However, Figure 11 shows white holes (no data) for all the 2015-2020 period. Please, explain how you have taking into account this absence of data in your Figure 12 (possibly, through extrapolation of data to the South pole?).

“Meteor M N2” has orbital inclination of about 98.77 ° and swath width of 1000 km. Thus, there are white circles (data gaps). We use only direct data, without any interpolation. We corrected the text in section 2.1 and 4.2 to explain this phenomena.

-In lines 682-683: The following text “and, finally, in 2020, essential ozone loss was observed mostly over the whole Northern Hemisphere”, describes a loss of ozone in the “whole Northern Hemisphere”, but this Hemisphere starts geographically at Equator. Consequently, consider to include the words “North part of the” after “whole” as follows: “North part of the whole Northern Hemisphere”.

Done

-Lines 824-825: This video (that this Reviewer had access through the Supplementary files submitted by the Remote Sensing journal) is a very interesting contribution to the presentation done by the Authors to their extensive and important work, but please verify Appendix B, since it was not possible to have access to Video 1 through the internet address given in line 825.

Done

Round 2

Reviewer 1 Report

Overall, I'm satisfied with the answers and the corrections made. I have only two questions:
(1) why do the authors plan to provide their ANN in binary format? Normally, the ANN can be provided in ASCII with scaled variables in the first section, combination of layers and coefficients in the second section, and the final result. In this form, it can be ported to any language whereas the binary file adds one level of complexity because not all binaries can be read the same way in different systems.
(2) I see some explanations of IKFS vs IASI differences in Section 4.1.2, but they are more of a general nature. Could you, please, be more specific and indicate, which dataset is more reliable (and why), the one coming from IASI or the one provided in this work?